# Congenital Diseases of DNA Replication: Clinical Phenotypes and Molecular Mechanisms

**DOI:** 10.3390/ijms22020911

**Published:** 2021-01-18

**Authors:** Megan Schmit, Anja-Katrin Bielinsky

**Affiliations:** Department of Biochemistry, Molecular Biology, and Biophysics, University of Minnesota, Minneapolis, MN 55455, USA; schm4097@umn.edu

**Keywords:** Meier-Gorlin syndrome, natural killer cell deficiency, X-linked pigmentary reticulate disorder, Van Esch-O’Driscoll disease, IMAGe syndrome, FILS syndrome, Rothmund-Thomson syndrome, Baller-Gerold syndrome, RAPADILINO

## Abstract

Deoxyribonucleic acid (DNA) replication can be divided into three major steps: initiation, elongation and termination. Each time a human cell divides, these steps must be reiteratively carried out. Disruption of DNA replication can lead to genomic instability, with the accumulation of point mutations or larger chromosomal anomalies such as rearrangements. While cancer is the most common class of disease associated with genomic instability, several congenital diseases with dysfunctional DNA replication give rise to similar DNA alterations. In this review, we discuss all congenital diseases that arise from pathogenic variants in essential replication genes across the spectrum of aberrant replisome assembly, origin activation and DNA synthesis. For each of these conditions, we describe their clinical phenotypes as well as molecular studies aimed at determining the functional mechanisms of disease, including the assessment of genomic stability. By comparing and contrasting these diseases, we hope to illuminate how the disruption of DNA replication at distinct steps affects human health in a surprisingly cell-type-specific manner.

## 1. Introduction

### 1.1. Replication Initiation

Accurate and efficient replication of our genome is critical to human health and development. Deoxyribonucleic acid (DNA) replication is composed of three main processes: initiation, elongation and termination. Initiation of DNA replication can be further divided into origin licensing and origin firing. Origin licensing begins at the end of mitosis (M phase) in the previous cell cycle and continues throughout gap 1 (G1) phase [1]. Licensing is the process by which a subset of replication components is assembled at an origin of replication in preparation for the initiation of DNA synthesis. Origin firing begins at the transition from G1-to-synthesis (S) phase of the cell cycle and this marks the time when double-stranded (ds) DNA begins to melt and separate into single-stranded (ss) DNA. Interestingly, more origins are licensed than are activated during the subsequent S phase [2]. Origins that are licensed, but do not fire, are termed dormant origins and serve as back-up origins if adjacent replication forks stall and cannot be reactivated [2]. In eukaryotic cells, origins are not all fired at the same time, with origins in more transcriptionally active regions characterized by open chromatin generally firing earlier than those in less active, heterochromatic regions [3]. Recent studies suggest that distinct replisome components operate at early and late replicating origins [4]. For instance, downstream neighbor of SON (DONSON), the homolog of the *Drosophila humpty dumpty* gene that is thought to promote replication fork stability may preferentially associate with early replicating origins [5,6]. Both origin licensing and firing are highly regulated processes to ensure that DNA is not re-replicated.

Origin licensing begins with the assembly of the origin recognition complex (ORC) on the DNA in an ATP-dependent manner [7,8]. ORC is a hetero-hexameric complex comprised of ORC1-6 subunits that encircle the DNA as an incomplete ring (Figure 1) [8]. ORC then recruits and binds cell division cycle 6 (CDC6) protein, which fills the gap in the ORC ring structure (Figure 1) [9]. Together, ORC and CDC6 recruit chromatin licensing and DNA replication factor 1 (CDT1) and one minichromosome maintenance 2-7 (MCM2-7) complex (Figure 1) [10,11]. MCM2-7 is another hetero-hexameric ring that functions as the core of the replicative helicase and CDT1 acts to hold this ring open at the interface between MCM2 and MCM5 (the MCM2/5 gate) [12]. It is thought that the ring-like structure of ORC and CDC6 enables loading of MCM2-7 onto dsDNA [9]. Additionally, ORC induces DNA bending, which may facilitate DNA interactions with the MCM2/5 gate in MCM2-7 [8]. After the first MCM2-7 ring is loaded, a second MCM2-7 complex is recruited in a cooperative manner [13,14,15]. The exact mechanism underlying the loading of the second MCM2-7 hexamer is debated; however, a quality control mechanism ensures that both hexamer complexes are loaded in a head-to-head orientation, bringing the N-termini of each hexamer together [13,16]. In a poorly understood process, the complexes can then bypass one another [16]. This is an additional regulatory step in replication initiation requiring subsequent origin firing factors to determine which origins of replication are activated [16].

Origin licensing and subsequent firing are tightly controlled by cyclin-dependent kinases (CDKs) to prevent re-replication after DNA synthesis is initiated (Figure 1) [1]. To be active, CDKs must bind a cyclin partner which also directs substrate specificity. Cyclin expression varies in a cell-cycle-dependent manner. In yeast, a single CDK partners with specific cyclins to regulate each cell cycle transition, whereas in humans, multiple cyclin/CDK pairs govern each phase of the cell cycle [1]. Both in yeast and humans, the active S phase CDKs prevent re-replication by phosphorylating components of the licensing machinery, including the ORC subunits, CDC6 and CDT1 [1]. This phosphorylation leads to protein degradation or exclusion from the nucleus, thus preventing licensing of new origins after the G1-to-S phase transition [1]. CDT1 is further controlled by Geminin (GMNN) [17]. GMNN is a protein that binds CDT1 and prevents its interaction with the MCM2-7 complex, thereby inhibiting origin licensing (Figure 1) [17]. GMNN levels fluctuate with cell cycle progression as well [18]. Specifically, GMNN is degraded during M phase and then slowly accumulates in S phase when its abundance is sufficiently high to sequester CDT1 [1,17,18].

As cells progress into S phase, firing factors assemble at replication origins to allow for the initiation of DNA unwinding and synthesis. The proteins that catalyze DNA replication are highly conserved across eukaryotes and much of the field’s understanding comes from studying model organisms, such as *Saccharomyces cerevisiae*. Many of these findings are consistent with observations in human cells and we will note where the two systems diverge. Utilizing purified proteins, Yeeles and coworkers demonstrated which firing factors were required for DNA replication initiation in an in vitro system [19]. These essential firing factors are DNA polymerase B II (Dpb11), synthetically lethal with Dpb11 2 (Sld2), Sld3, Sld7, cell division cycle 45 (CDC45), go-ichi-ni-san (GINS) complex, DNA polymerase epsilon (pol ε) and MCM10 [19]. *Sld3*, *Sld7*, *Sld2* and *Dpb11* do not have homologous genes in humans; however, they have orthologs or analogs: *Treslin*, *Mdm2-binding protein* (*MTBP*), *ATP-dependent DNA helicase Q4* (*RECQL4*) *and DNA topoisomerase II binding protein 1* (*TopBP1*), respectively (Figure 1) [20,21,22,23,24].

The recruitment of these factors is regulated by S phase CDK and Dbf4-dependent kinase (DDK) [1,19,25,26]. As cells transition into S phase, DDK accumulates and phosphorylates MCM2, 4 and 6 [1,19,26]. These phosphorylations likely lead to a conformational change, facilitating the recruitment of CDC45 and Sld3/Sld7 [19,27]. Sld3/Sld7 are needed to stabilize CDC45 binding to MCM2-7 until the GINS complex is recruited with Sld3, Sld2, Dpb11 and pol ε following S-CDK phosphorylation of Sld3 and Sld2 [19,28,29]. Similarly, in humans, TopBP1 and MTBP interact with Treslin in a CDK-dependent manner and are required for CDC45 recruitment and stabilization [23,24,30,31]. Unlike the role that Sld2 plays in GINS recruitment in yeast, RECQL4 has not been shown to interact with GINS but is required for replication initiation, possibly by stabilizing the CMG complex after GINS loading [22,31,32]. Furthermore, whereas Sld3, Sld7, Sld2 and Dpb11 are thought to dissociate from the replication fork after DNA melting, in humans, TopBP1 continues to travel with the elongating replication fork (Figure 1) [33].

Both CDC45 and the GINS complex are helicase coactivators. With the loading of the GINS complex, the CDC45-Mcm2-7-GINS (CMG) helicase is completely assembled, and a conformational change occurs in which each CMG complex encircles ssDNA by extruding the lagging strand template [34,35]. It remains unclear if the same MCM2/5 gate that facilitates dsDNA loading also serves as the ssDNA gate for lagging strand template extrusion [34,35,36]. What is known is that CDC45 and the GINS complex bind across the MCM2/5 gate and may be needed to facilitate ssDNA extrusion or prevent leading strand template escape [34,35,36]. MCM10 is required for significant ssDNA generation and subsequent replication protein A (RPA) binding (Figure 1) [37,38,39]. MCM10′s role in generating ssDNA during origin firing may be to enable the head-to-head oriented CMG complexes to separate and slide past each other to initiate unwinding [40]. As DNA replication progresses, topoisomerase II eases super coiling caused by strand unwinding [19]. With origin firing complete, the two nascent forks move away from the origin and MCM10 remains associated with CMG during elongation [41].

### 1.2. Replication Elongation

Elongation is the process in which DNA polymerases catalyze the formation of the phosphodiester bond between the 3′ hydroxyl group of the growing strand and the 5′ phosphate group of the incoming nucleotide while using the parental strand as a template. DNA polymerases add nucleotides to a primed template in the 5′ to 3′ direction [42]. As the CMG helicase unwinds parental DNA, the strand with 3′ to 5′ polarity serves as the leading stand template and can be duplicated continuously. The other strand, the lagging strand template, is copied in a discontinuous manner by the repeated synthesis of Okazaki fragments. For DNA synthesis to begin, MCM10 and acidic nucleoplasmic DNA-binding protein 1 (AND-1) recruit polymerase alpha-primase (pol α-primase), which synthesizes a short ribonucleic acid (RNA)-DNA primer (Figure 1) [43,44,45]. On the lagging strand, replication factor C (RFC) recognizes the ds-to-ss junction at the RNA-DNA primer:template with a recessed 3′ end as a substrate onto which it loads proliferating cell nuclear antigen (PCNA), the tether for DNA polymerase delta (pol δ) (Figure 1) [19,46,47]. Pol ε is loaded onto DNA before origin firing and is already present at the replication fork to take over synthesis of the leading strand [43]. Further processing of Okazaki fragments is needed to remove the RNA primers. This occurs when pol δ, aided by PCNA, encounters the previous Okazaki fragment and displaces the RNA portion to generate a 5′ flap [48,49,50]. This displaced 5′ flap can then be cleaved by flap endonuclease 1 (FEN1) or DNA replication helicase/nuclease 2 (DNA2) depending on the length of the 5′ flap [48,49,51,52]. The flap cleavage results in a DNA nick that is sealed by DNA ligase 1 [53].

### 1.3. Replication Termination

Termination of replication occurs when two forks converge. One proposed model for termination has CMG helicases pass each other and transition from encircling ssDNA to dsDNA when they reach the 5′ end of the previously synthesized Okazaki fragment [54,55]. Synthesis of the leading strand is completed when pol ε reaches this fragment shortly thereafter. The replisome then needs to be disassembled. CRL2^Lrr1^ (a Cul2-based RING E3 ubiquitin ligase) polyubiquitylates MCM7, leading to CMG unloading by CDC48/p97 segregase [56,57]. The CMG components are not degraded but rather recycled [58]. Currently, there are no known congenital diseases caused by disruption of replication termination.

As summarized above, DNA replication is a multistep process involving a multitude of protein complexes. In this review, we describe inherited genetic diseases caused by defects of proteins essential for DNA replication. For each disease, we describe the clinical presentation, report associated genetic mutations and discuss studies of their pathophysiologies. As we explore diseases caused by defects in the core replication machinery, we highlight both expected commonalities, such as severe growth defects, as well as tissue-specific abnormalities. Interestingly, mutations in individual replication factors can cause glaringly unique and disease-specific symptoms. The field of DNA replication has yet to fully understand why mutations in different DNA replication proteins cause these diverse symptoms. We propose that different cell types have specific thresholds for replication factors that govern proper differentiation. Specifically, each disease and associated genetic defect can be broadly grouped into two categories that display some homogeneity in symptomology—defects in origin licensing or defects in origin firing/elongation.

## 2. Diseases

### 2.1. Meier-Gorlin Syndrome Is Caused by Defects in Origin Licensing

Meier-Gorlin syndrome (MGS) is an osteodysplastic syndrome that causes primordial proportional dwarfism. In addition to growth retardation, microtia (small ears) and aplastic or hypoplastic patella comprise the three core clinical findings in MGS (Table 1) [59]. The first patient was described by Meier and coworkers in 1959, with the second described in 1975 by Gorlin et al. [60]. Several other case reports soon followed under descriptive titles such as small ear syndrome or ear, patellae, short stature syndrome [61,62,63]. In 1994, the name MGS was proposed to identify patients with the triad of symptoms described above and separate them from other primordial dwarfism disorders [64]. As MGS is characterized by osteodysplasia, early reports by Gorlin hypothesized that the genetic cause lay in genes that direct bone development [65,66]. However, it was not until 2011 that the genetic cause of this syndrome was determined to be mutations in various origin licensing factors including *ORC1*, *ORC4*, *ORC6*, *CDT1* and *CDC6* (Figure 2 and Figure 3) [67,68]. Later studies have found causative mutations in *CDC45*, *MCM5*, *GMNN* and *DONSON* (Figure 4 and Figure 5; for complete list of mutations including DNA and protein change, see Appendix A) [5,69,70,71].

Growth retardation in MGS can range from mild to severe. *ORC1* and *ORC4* mutations cause the most severe growth defects [72,73,143,144]. In addition to short stature, microtia and hypo/aplastic patellae, MGS patients have distinctive facial features including microstomia, full lips, retrognathia and micrognathia (Table 1) [72,75]. Intellect appears to be normal although there may be delayed motor or speech development [72]. Feeding problems are prevalent in approximately 80% of patients and respiratory tract anomalies occur in nearly half of MGS patients (Table 1 and Appendix A) [72]. Additionally, while only 7% of MGS patients have a congenital cardiac anomaly, it is advised that cardiac screening be undertaken if a diagnosis of MGS is made (Appendix A) [72]. As children with MGS grow into adulthood, they can develop conductive hearing loss due to microtia, arthritis in the knee due to absent patella and pain due to general joint hypermobility (Appendix A) [72,144]. Fertility of patients with MGS remains unclear and may be decreased in males born with cryptorchidism and in females with small uteri and polycystic ovaries [72,143]. Additionally, there is abnormal secondary sex development with sparce axillary hair in both sexes and hypoplastic mammary tissue in women (Appendix A) [144].

In 2013, Stiff and coworkers assessed immortalized fibroblasts from an MGS patient with *ORC1* mutations as well as lymphoblastoid cell lines derived from patients with *ORC1*, *ORC4*, *ORC6*, *CDT1* and *CDC6* mutations to understand the etiology of MGS [145]. Epstein–Barr virus (EBV) replication, which requires highly efficient origin licensing, was reduced in these patient-derived cells [74]. In 2017, the Meng laboratory demonstrated that zebrafish models of MGS exhibited delayed S phase progression and increased apoptosis [146]. Furthermore, only some tissue types were affected, suggesting that the effects are cell-type-specific [146]. Similarly, several *CDT1* mutations identified in MGS patients slowed origin licensing and delayed entry into the S phase [10]. Interestingly, one hypermorphic allele caused re-replication, replication stress and impaired proliferation [10]. Studies of fibroblasts from ORC1 patients also demonstrated delayed S phase entry and progression due to impaired origin licensing [146]. Additional biochemical studies of *Drosophila* ORC6 have demonstrated that mutation of Y232 resides within a highly conserved helix that facilitates interactions with ORC3 [147]. ORC6 plays an important role in recruiting ORC to DNA, and without this interaction, ORC complex binding to DNA and subsequent MCM2-7 loading is diminished [147]. Similarly, disruption of the bromo-adjacent homology domain in the N-terminus of ORC1 (as in c.266T > A, c.314G > A, and c.380A > G), which mediates its interaction with histone H4 dimethylated at lysine 20, disrupts ORC assembly on DNA [148,149].

Stiff and colleagues noted that other primordial dwarfism syndromes such as Seckel syndrome have been linked to defects in ciliogenesis [145,150]. Furthermore, Seckel syndrome (SS) can be caused by mutations in ataxia-telangiectasia and Rad3-related protein (ATR), a kinase critical for activation of the DNA damage checkpoint [151]. The related functions of MGS pathogenic variants in DNA replication and SS pathogenic variants in sensing replication stress could indicate a common etiology for these two primordial dwarfisms. When fibroblasts were depleted for ORC1, ORC4, ORC6, CDT1 and CDC6, primary cilia formation was impaired, suggesting that MGS symptoms might constitute a ciliopathy [145]. However, MGS patients do not have the cognitive impairment, abnormal neurogenesis or renal defects that are often present in primary ciliopathies [152,153]. Thus, it remains an open question as to whether a direct link exists between MGS and aberrant cilia formation. Following the original studies identifying mutations in *ORC*, *CDC6* and *CDT1* as causing MGS, additional reports revealed *MCM5, CDC45*, *GMNN* and *DONSON* variants in MGS patients (Figure 4 and Figure 5) [5,69,70,71,76,77,78]. One patient was identified with a compound heterozygous mutation in *MCM5* resulting in a reduction in MCM5 and MCM2 on chromatin, consistent with an inability to stably load MCM2-7 onto chromatin (Figure 4 and Appendix A) [70]. Moreover, fifteen patients with biallelic *CDC45* mutations, including both compound heterozygous or homozygous variants, which conferred varying degrees of protein dysfunction and depletion were described (Figure 4 and Appendix A) [69,77]. Interestingly, *CDC45* MGS patients, in addition to the traditional triad of MGS symptoms, were more likely to have craniosynostosis and anorectal malformations (Table 1 and Appendix A) [69,77]. Patients carrying heterozygous *GMNN* mutations are afflicted by an autosomal dominant form of MGS (Figure 5 and Appendix A) [71]. All mutations produce a N-terminally truncated protein using an alternate start codon [71]. The use of this later start codon resulted in a highly stabilized protein that decreased DNA replication due to increased CDT1 inhibition [18,71,154]. It remains unclear whether mutations in *DONSON* represent true MGS or represent a related syndrome [5,77,153]. Patients with *DONSON* mutations present with primordial dwarfism, but not all have the classic triad of MGS symptoms [5,76,78]. Molecular studies of DONSON demonstrated a function in DNA replication fork stabilization rather than DNA replication initiation [5]. However, due to the lack of patient samples and appropriate models, it remains unclear whether DONSON mutations that phenotypically mimic MGS also have origin licensing defects. When taking into account all the genes found to be mutated in MGS patients, it seems clear that the disease phenotype is due to defects in DNA replication, although the connections to impaired cilia formation cannot be ruled out as a contributing factor.

### 2.2. MCM2 Deficiency and Familial Deafness

The Liu laboratory identified a missense variant in *MCM2* that segregated in an autosomal dominant manner with sensorineural hearing loss in a large family cohort (Figure 6) [142]. This missense variant changes highly conserved R44C (c.130C > T) (Figure 6) [142]. Hearing loss in this family was variable for onset, bilateral and progressive [93]. *MCM2* was expressed at low levels in various parts of the cochlea [142]. In hair cells specifically, most of the MCM2 protein was found in the cytoplasm, consistent with the non-proliferative state of these cells [142]. As previously reported, overexpression of MCM2 in HEK293 cells resulted in increased apoptosis [142]. Apoptosis further increased with overexpression of the R44C variant [142]. This suggests that accumulation of the R44C variant in the cytoplasm of hair cells may be sufficient to induce apoptosis, a known mechanism for hearing loss [142,155].

### 2.3. Natural Killer Cell Deficiency Is Caused by Defects in Origin Licensing or Initiation Factors and Is Associated with Genome Instability

The human immune system is broadly divided into the innate and the adaptive immune system. The adaptive immune system has high specificity for pathogens and includes humoral components such as T lymphocytes (T cells), which recognize pathogens through their diverse T cell receptors, and the antibodies produced by B lymphocytes (B cells). Conversely, the innate immune system is not specific for individual pathogens and includes a third class of lymphocytes, the natural killer (NK) cells [156]. NK cells have multiple activating receptors including cluster of differentiation (CD)16, a receptor for the fragment crystallizable region (Fc) of antibodies [156]. They also express inhibitory receptors called killer immunoglobulin-like receptors (KIRs) that recognize major histocompatibility complex (MHC) class I molecules [156]. KIRs equip NK cells to target and kill virally infected or tumor cells that elude T cell recognition due to the downregulation of MHC molecules [156]. In addition to their cytotoxic activity, NK cells have important roles in immune-modulation through the secretion of cytokines such as interferon-γ and tumor necrosis factor-α [156]. NK cells develop from the same common lymphoid progenitor as B and T cells and go through a multistep process of maturation primarily in the bone marrow [156]. One model of NK cell development divides the process into five stages, with stage 5 being the fully mature cytotoxic NK cell [156,157,158,159]. Both stage 4 and 5 NK cells are present in the peripheral blood. NK cells range between 3 and 30% of all lymphocytes within the peripheral blood and stage 5 NK cells are nine times more abundant than stage 4 NK cells [160,161,162].

Natural killer cell deficiency (NKD) is a rare primary immune deficiency affecting specifically NK cells. NKD results in increased susceptibility to common pathogens and has severe effects on health and lifespan. NKD is further subdivided into functional (FNKD) and classical NKD (CNKD). FNKD is the presence of normal levels of NK cells in peripheral blood that are not able to induce cell lysis. CNKD is caused by a lack of NK cells in peripheral blood to less than 1% of total lymphocytes [163]. CNKD can be caused by pathogenic variants in DNA replication factor genes, suggesting an unexplored connection between NK cell development and DNA replication. The genes affected in individuals with CNKD are *MCM4*, *GINS1* and *MCM10*, suggesting that the disease is caused by defects in origin licensing and/or firing (Figure 7) [84,160,163].

The *MCM4* CNKD cohort is the largest and arises in a genetically isolated Irish traveler population [79,80,81]. In addition to CNKD, affected individuals in this cohort have adrenal insufficiency, pre- and post-natal growth retardation and unique facial features (Table 1) [79,80,81,82,164]. When NK cell populations were analyzed from these patients, a dramatic lack of stage 5 NK cells was noted [80]. Whereas the amount of stage 4 cells within the peripheral blood was normal, these cells were unable to be expanded ex vivo with cytokine stimulation and underwent spontaneous apoptosis likely due to increased chromosome breakage [80]. Within the *MCM4* cohort, CNKD is inherited in an autosomal recessive manner with the gene defect ablating a splice site in an early exon generating a premature stop codon (Figure 7 and Appendix A) [79,80,81]. Despite this splice site mutation, mature *MCM4* mRNA can be detected at levels comparable to healthy cells [80]. Furthermore, two proteins smaller than full-length MCM4 were expressed, corresponding to two naturally occurring isoforms of MCM4 that are initiated at two separate start sites downstream of the splice site mutation and resulting premature stop codon [80]. These two MCM4 isoforms did not appear to affect MCM2-7 stability on chromatin; however, they were not sufficient to allow normal cell cycle progression in immortalized patient fibroblasts which arrested in G2/M phase. This indicates that the full-length MCM4 is necessary for normal origin firing and elongation but not for origin licensing. Patient cells also had increased DNA content, demonstrating that the N-terminal region of full-length MCM4 is required for the appropriate regulation of DNA synthesis and prevention of re-replication [80]. When treated with aphidicolin, a DNA replication inhibitor, these cells exhibited increased chromosomal breakage [80]. Similarly, cells obtained from mice with a hypomorphic *MCM4* allele also had increased chromosome breaks after aphidicolin treatment [165]. Regions that undergo breaks with aphidicolin treatment often correspond to common fragile sites; however, follow-up studies to identify the chromosomal location of these breaks in MCM4 patients were not described [80,166]. The increase in genomic instability seen with a hypomorphic *MCM4* allele led to an increased cancer rate in the mouse [165]. It is unclear if *MCM4* pathogenic variants lead to cancer in humans as well. Some reports of cancers have been made within the *MCM4* cohort, but all of them could be attributable to viral transformation secondary to NK cell depletion [167]. At the very least, this increase in DNA content and DNA breaks with replication stress indicates that these hypomorphic MCM4 proteins are unable to perform efficient DNA replication and prevent re-replication.

Unlike the *MCM4* cohort, the *GINS1* cohort is composed of five individuals from four unrelated families and presented with a slightly different phenotype [83]. These individuals have pre- and post-natal growth retardation and unique facial characteristics, eczema and lymphadenopathy (Table 1 and Appendix A) [83]. They do not have adrenal insufficiency [83]. Other clinical presentations that were reported include features of premature aging, hypothyroidism, protein-losing enteropathy [83], diffuse osteopenia and low amounts of CD8^+^ T cells [83]. All patients had a near complete lack of both stage 4 and stage 5 NK cells, as well as chronic neutropenia. Interestingly, upon stimulation, as in bacterial infection, neutrophil numbers could be restored [83]. Each patient had a compound heterozygous variant of the *GINS1* gene (Figure 7 and Appendix A) [83]. Among the five patients, four unique variants, two missense mutations (c.247C > T;p.R83C and c.455G > A;p.C152Y) and two splice site mutations affecting the 5′ untranslated region (UTR) (c.−48C > G and c.−60A > G) were identified (Figure 7 an Appendix A) [83]. The 5′ UTR mutations resulted in the use of downstream start codons, causing N-terminal truncation of the protein; however, they did not cause complete aberrant splicing and each patient expressed low levels (<10%) of full-length *GINS1* transcript [83]. Accordingly, formation of the GINS complex was compromised, but not ablated. The C152Y variant was able to form the GINS complex, but not the CMG complex [83]. Similar to *MCM4* patient-derived cells, fibroblasts from *GINS1* patients arrested in the G2/M phase of the cell cycle and displayed increased DNA content [83]. They also exhibited decreased proliferation, increased basal DNA damage and increased β-galactosidase activity, a marker for cellular senescence [83]. DNA fiber analysis revealed fewer initiation events, increased fork stalling and faster replication fork speed compared to control cells, indicative of replication stress driven by initiation defects and fork instability [83].

We recently described a new NK cell deficiency patient presenting with severe cytomegalovirus infection [84]. The patient was not reported to have growth retardation, adrenal insufficiency or dysmorphic facial features [84]. This patient had an over representation of stage 4 and a severe reduction in stage 5 NK cells with mildly decreased T and B cell counts [84]. Similar to the *GINS1* cohort, this patient was a compound heterozygote with pathogenic variants in *MCM10* (Figure 7 and Appendix A) [84]. One allele contained a nonsense mutation (c.1744C > T;p.R582X) that would truncate the protein prior to the nuclear localization signal, making it functionally a null allele [84]. As a truncated protein could not be detected, the transcript is likely degraded by nonsense mediated decay [84]. The second allele carries a missense mutation (c.1267C > T;p.R426C), changing a highly conserved arginine to cysteine (Figure 7 and Appendix A) [84]. Immortalized patient-derived fibroblasts exhibited increased DNA damage and abnormal cell cycle progression with an increase in cells in the S phase [84]. In model cell lines, each mutation separately resulted in shortened telomeres [85]. Strikingly, MCM10 deficiency caused not only increased inter-origin distance due to decreased origin firing, but telomere combing revealed an increased frequency of unreplicated and partially replicated telomeres [85]. Moreover, detailed karyotype analysis revealed increased chromosomal translocations, mostly originating from breaks at common fragile sites, which are known to be origin-poor [168]. Thus, telomeres and common fragile sites are particularly affected by reduced origin firing. Based on these observations, we propose that NK cell deficiency may actually constitute a telomeropathy, causing premature senescence in the NK cell lineage and providing an explanation for the lack of mature, stage 5 cells [85]. This hypothesis is supported by a case in which an individual with a *regulator of telomere length 1* (*RTEL1*) pathogenic variant presented with NKD [169]. Genetic changes in *RTEL1* have been primarily associated with Hoyeraal–Hreidarsson syndrome, a severe telomeropathy [170]. Furthermore, many other telomeropathies have low NK cell counts within the context of global cytopenia [171,172,173]. Peripheral NK cells are known to have shorter telomeres than T or B cells and a lower replicative potential than other lymphocytes [158,174,175]. Therefore, defects in *MCM10* may predispose NK cells to advanced telomere erosion and premature senescence.

In addition to the NKD caused by pathogenic variants in *MCM10*, we also recently described a family with a history of intra-uterine fetal death due to reduced cardiac function and fetal hydrops [85]. These fetuses had biallelic pathogenic variants c.236delG and c.764 + 5G > A in *MCM10* (Figure 7) [85]. One of these variants (c.236delG, p.G79EfsX6) disrupts exon 3 and is essentially a null allele [85]. The other variant reduces the splicing efficiency of exon 6, resulting in low levels of wild-type *MCM10* expression [85]. Together, these two variants result in a greater reduction in MCM10 function than the compound heterozygous variants in the NKD patient [85]. On postmortem examination, all three fetuses had hypoplastic spleens and thymuses, suggesting that immune cell development was affected [85]. The cardiac dysfunction leading to fetal death was diagnosed as restrictive cardiomyopathy. Interestingly, cardiomyopathy has been associated with telomere erosion in cardiomyocytes, further supporting a role for telomere dysregulation in clinical syndromes due to MCM10 deficiency [85,176,177].

Within the context of NKD caused by replication defects, we will discuss a related disease, X-linked pigmentary reticulate disorder (XLPRD). This is another rare syndrome with recurrent infections caused by pathogenic variants in *POLA1* which leads to reduced pol α expression (Figure 8) [86]. In addition to recurrent infections, affected individuals have hypogonadism, hyperpigmentation, corneal dystrophy, photophobia, unique facial features and multiorgan inflammation due to the role of pol α in modulating the interferon response [86,87,88,89,90]. Recurrent infections are due to reduced NK cell numbers and reduced cytotoxicity of peripheral NK cells, although the NK cells in these individuals do not stay consistently low enough to make the diagnosis of CNKD [90]. Interestingly, NK cells in these patients appear to also have impaired cytotoxic function, conceptually making the immune deficiency in these individuals a combination of CNKD and FNKD [90]. Surprisingly, immortalized patient fibroblasts did not have any proliferation or cell cycle defects, arguing that pol α expression remained above the necessary threshold to support normal DNA replication in vitro [90]. Whether this is true during the replicative cycles as NK cells undergo development in vivo is still an open question.

### 2.4. Multiple Pol α Subunits Are Linked to Human Disease

In addition to XLPRD, Van Esch-O’Driscoll disease has been linked to pathogenic variants of *POLA1* in nine individuals belonging to five unrelated families (Figure 8 and Appendix A) [91,92]. This disease is characterized by intellectual disability, growth retardation, microcephaly and hypogonadism (Table 1) [91]. Additional notable features that occur in some of these patients include hypotonia, behavioral challenges, cerebral atrophy, congenital heart defect, facial dysmorphism, recurrent infections, esophageal atresia and epilepsy (Appendix A) [91,92]. All affected individuals are male, consistent with a X-linked recessive mode of inheritance [91]. Each family within this cohort has a different pathogenic variant (Figure 8 and Appendix A) [91]. As in individuals with XLPRD, these mutations can cause reduced expression of *POLA1* [91]. However, substitution of highly conserved residues, such as G110R, I79S, P1381L, and deletion of amino acids 149-169 directly affect protein function (Figure 8 and Appendix A) [91]. DNA combing in lymphoblastoid cell lines from these patients revealed diminished replication initiation events, increased inter-origin distance and increased fork asymmetry, consistent with impaired origin firing and replication fork instability [91]. Moreover, cell lines from multiple families displayed increased sensitivity to the replication inhibitor hydroxyurea, consistent with the notion that defects in pol α increase sensitivity to replication stress [91]. In zebrafish, *POLA1* is highly expressed in the brain both pre- and post-birth, explaining why normal brain function can be impaired in humans [91]. Pathogenic variants in Van Esch-O’Driscoll disease could have more severe effects on growth and brain development because the compound heterozygous mutations lead to exacerbated loss of function, but it is not clear why global inflammation and recurrent infections (as seen in XLPRD) are not a consistent component of the disease phenotype. More work is needed to understand how these two diseases progress during development.

Pathogenic variants in *PRIM1*, the catalytic subunit of DNA primase which is required for the initiation of leading and lagging strand synthesis, have been identified in four families afflicted with microcephalic dwarfism (Figure 8) [93]. In addition, these individuals have unique facial appearance, reduced subcutaneous fat, hypothyroidism, cardiac abnormalities, hepatic dysfunction indicative of chronic liver inflammation and immune dysfunction [93]. The microcephaly is accompanied by abnormal cerebral cortex structure and severe developmental delay [93]. The immune dysfunction includes agammaglobulinemia and lymphopenia, with intermittent anemia and thrombocytopenia. The severe immune dysfunction in these individuals appears to have contributed to early death in five out of six individuals [93]. The most common pathogenic variant, c.638 + 36C > G, inherited biallelically in three of the four families, results in mis-splicing of the *PRIM1* transcript and subsequent nonsense mediated decay (Figure 8) [93]. As expected with decreased *PRIM1* expression, patient-derived cells showed increased doubling time, reduced DNA synthesis, increased levels of DNA damage and delayed progression through the G1 phase of the cell cycle [93]. Several of the clinical features as well as molecular defects overlap with diseases resulting from pathogenic variants in other polymerase subunits (discussed below), as well as with Van Esch-O’Driscoll disease and XLPRD (Table 1). This suggests that perturbations of either lagging or leading strand synthesis can have similar effects on disease outcomes.

### 2.5. Pol ε Dysfunction Causes Immunodeficiency

Pol ε is composed of four subunits, POLE1, POLE2, POLE3 and POLE4. Pol ε functions as the major leading strand polymerase and in DNA repair [42]. Pathogenic variants of *POLE1* can cause non-syndromic cancer predisposition (not discussed here). Additionally, mutations in POLE1 can cause intrauterine growth restriction, metaphyseal dysplasia, adrenal hypoplasia congenita and genitourinary abnormalities (IMAGe) syndrome and facial dysmorphism, immunodeficiency, livedo and short stature (FILS) syndrome [94,97,178]. Most individuals with IMAGe syndrome carry pathogenic variants in cyclin-dependent kinase inhibitor 1C (CDKN1C/p57), a negative regulator of cell proliferation [97]. Interestingly, loss-of-function variants of CDKN1C lead to an overgrowth syndrome, whereas gain-of-function variants localizing to the PCNA binding domain of CDKN1C cause IMAGe syndrome [179]. *POLE1* was identified as an additional gene involved in IMAGe syndrome in 2018 by whole-genome sequencing of a cohort with microcephalic primordial dwarfism [97]. Fifteen individuals with biallelic mutations in *POLE1* and features of IMAGe syndrome were identified in this study (Figure 9) [132]. Each individual carried one allele with the same intronic variant that resulted in abnormal splicing of exon 15 (Figure 9) [97]. Whereas some wild-type transcript could still be produced from this allele, the majority contained a 47 bp insertion, leading to a premature stop codon in the catalytic domain [97]. The second allele in these individuals had various loss-of-function mutations and patient-derived fibroblasts displayed impaired S phase progression [97]. Phenotypically, these individuals not only had features of IMAGe syndrome but also had immunodeficiency and common facial dysmorphism [97]. The immunodeficiency seen here was variable with low T, B or NK cell counts, but the majority suffered increased infection rates [97]. Lastly, these individuals may also have increased likelihood of cancer as two individuals in this cohort developed blood cancers [97].

In addition to IMAGe syndrome, reduced expression of POLE1 in a large consanguineous family caused FILS [94]. These individuals carried a mutation of intron 34, resulting in abnormal splicing and a significant reduction in the expression of full-length *POLE1*, as well as a corresponding decrease in POLE2 protein (Figure 9) [94]. The reduction in pol ε subunits unsurprisingly led to decreased proliferation of patient cells in vitro [94]. The de Saint Basile laboratory determined POLE1 levels in multiple tissues and identified that tissues with high expression, including chondrocytes and lymphocytes, corresponded with the clinical phenotypes [94]. Similar to the cohort of IMAGe patients, the FILS cohort also has variable immune cell phenotypes with reduced B or NK cells [94]. The most striking immune phenotype was the reduction of memory B cells and IgM and IgG levels [94]. Two additional case reports from unrelated families have been described with similar symptoms to FILS syndrome [95,96]. Surprisingly, one of these patients had the same intronic mutation as the original family but with slightly different presentation [96]. The third case report described a homozygous substitution of R34C (c.100C > T), which lies outside the exonuclease and polymerase domains [95]. These additional patients had short stature, immunodeficiency, facial dysmorphism with slight differences from the original description and poikiloderma rather than livedo. Additionally, they presented with developmental delays and cryptorchidism, as might be seen in an individual with IMAGe syndrome.

POLE1 is not the only pol ε subunit that plays a role in human disease. A patient with immunodeficiency, facial dysmorphism, short stature and autoimmunity has been reported with homozygous splice site mutations in intron 13 of *POLE2* (Figure 9) [98]. This gene variant results in either the use of a cryptic splice site +3 nucleotides into exon 14, or exon 14 skipping [98]. Whereas this mutation did not appear to affect the overall expression level of *POLE2*, use of the cryptic splice site leads to deletion of the highly conserved S359 [98]. Patient-derived cells accumulated in the G2/M phase of the cell cycle and harbored chromosome duplications rather than rearrangements [98]. The immune phenotype in this patient appeared quite severe, with the near absence of mature B cells as well as reduced NK and T cells [98]. Thus, disruption of pol ε function causes a wide range of phenotypic outcomes, even when the defects occur in the same subunit as in IMAGe and FILS. What is particularly striking is that these symptoms overlap with diseases affecting replication origin licensing factors (MGS and skeletal anomalies), firing factors (NKD and immunodeficiency) and repair factors (RECQL4 and cancer predisposition/skin phenotypes). This is generally in agreement with the multiple roles that pol ε plays in maintaining genomic stability.

### 2.6. Pol δ Associated Progeroid Syndrome and Immune Deficiency

In 2010, Shastry and colleagues described a group of patients with mandibular hypoplasia, deafness, progeroid features (MDP) including lipodystrophy with the loss of subcutaneous fat and elevated visceral fat [99]. These individuals had some phenotypic overlap with mandibuloacral dysplasia (MAD). MAD is caused by pathogenic variants in *LMNA*, which encodes lamin A/C components of the nuclear cytoskeleton and *ZMPSTE24*, a zinc metalloproteinase required in the formation of mature lamin A [180]. Symptoms shared by these two syndromes are mandibular hypoplasia, bird-like face or pinched nose, short stature, joint contractures and scleroderma of the skin [99,180]. MDP patients do not have the skeletal abnormalities seen in MAD patients, nor do they have pathogenic variants of *LMNA* or *ZMPSTE24* [99,180]. Unique features of MPD include sensorineural hearing loss in childhood, hypogonadism and undescended testes in affected males and dry eyes [99,100,180]. Additionally, MDP patients have longer survival than MAD patients, although they have multiple comorbidities consistent with a progeroid syndrome, including dyslipidemia, cardiovascular disease and hyperglycemia [99,100,101,102,103,104,105,106,107]. Interestingly, MDP is caused by a *de novo* monoallelic pathogenic variant in *POLD1* which encodes the catalytic subunit of pol δ (Figure 10 and Appendix A) [101]. This pathogenic variant (c.1812_1814delCTC) results in the loss of S605 within motif A of POLD1 [101]. Motif A is responsible for orienting the incoming nucleotide to catalyze the phosphodiester bond and deletion of S605 ablates the polymerase activity of the enzyme [101]. The stability of this variant protein suggested that it may outcompete wild-type pol δ produced from the second normal allele in these individuals, making this a dominant negative mutation [101]. Exonuclease activity of the mutant pol δ was maintained, albeit at a slightly lower level than the wild type [101]. This is noteworthy because other pathogenic variants of pol δ affecting exonuclease activity have been associated with cancer predisposition syndromes [101]. In total, 21 individuals have been reported with MDP; 16 of these carried the S605del mutation [99,100,101,102,103,104,105,106,107]. One of the additional pathogenic variants is a R507C (c.1519C > T) substitution [104]. This amino acid is in the exonuclease domain, but not in the active site, and therefore is also not likely to increase cancer risk [104,105]. This variant is not predicted to impact polymerase activity directly, but may play a role in maintaining pol δ on chromatin during DNA elongation [105]. A third pathogenic variant is a I1070N (c.3209T > A) substitution reported in a single individual [106]. This mutation is predicted to severely impair protein folding and results in a severe clinical phenotype with fast onset of progeroid features and recurrent infections [106]. A fourth pathogenic variant, E1067K (c.3199G > A), resides in the zinc finger domain and was identified in two related individuals [107]. This variant resulted in abnormal nuclear structure, although to a lesser extent than seen with pathogenic changes in *LMNA* [107]. Interestingly, some MPD patients were originally diagnosed with Werner syndrome, suggesting phenotypic overlap and variable phenotypes [105]. Pol δ interacts with the Werner helicase and Werner syndrome shares many characteristics with MPD, including a bird-like face, loss of adipose tissue, joint contractures, scleroderma, hypogonadism and insulin resistance [101,105]. Werner helicase enhances pol δ activity, particularly at difficult-to-replicate secondary structures within common fragile sites, providing a rationale for the overlapping clinical phenotypes [181,182,183].

Pathogenic variants in pol δ disrupting *POLD1* and *POLD2* have also been linked to immunodeficiency, similar to diseases seen with defects in other polymerase subunits (Figure 10) [90,93,94,96,97]. Affected individuals had severe lymphopenia of NK cells and CD4^+^ T cells, reduced CD8+ T cells, agammaglobulinemia, abnormal distribution of progenitor cells in the bone marrow or B cell lymphopenia [108,109,110]. Studies with patient-derived cells demonstrated impaired T cell development and proliferation [108,109,110]. T and NK cells exhibited increased spontaneous DNA damage and NK cells were defective in their response to X-ray irradiation [109]. DNA combing of patient fibroblasts showed reduced origin firing with a compensatory increase in fork speed [108]. Additionally, the S phase checkpoint was activated and an increase in p53 binding protein 1 (53BP1) nuclear bodies was observed in the G1 phase of the cell cycle, indicative of replication stress and under-replicated DNA [108,109]. Additional symptoms included short stature, intellectual disability and hearing loss. Although there is minor overlap of symptoms with MDP patients, it is unclear why shared features are limited. One clue may lie in the fact that pathogenic variants resulting in immunodeficiency are autosomal recessive and inherited from healthy parents, whereas those in MDP are *de novo* autosomal dominant mutations, suggesting that the recessive alleles have a less severe impact on pol δ function than MDP alleles.

### 2.7. PCNA Associated Human Disease

Four individuals from a large consanguineous family were identified with neurodegeneration, short stature, prelingual sensorineural hearing loss, premature aging, telangiectasias, intellectual disability, photophobia and photosensitivity [111]. The neurodegeneration appeared progressive in three of the patients at the time of study, with increasing difficulty in swallowing [111]. One patient had particularly severe loss of muscle control, resulting in the loss of speech and the use of a wheelchair [156]. Baple and colleagues identified a S228I (c.683G > T) pathogenic variant in *PCNA* (Figure 11) [111]. S228 is near the outer surface of PCNA, so it likely has no effect on DNA binding. Indeed, analysis of patient-derived cells assessed by BrdU incorporation demonstrated that this pathogenic variant of *PCNA* did not cause defects in DNA synthesis [111]. Interestingly, the patient phenotypes have similarities with xeroderma pigmentosum (XP), ataxia telangiectasia (AT) and cockayne syndrome (CS), which are all caused by defects in the DNA damage response and DNA repair [111]. Specifically, XP and CS are caused by defects in nucleotide excision repair (NER) [184]. Considering the phenotypic overlap and the critical role PCNA plays in NER, it is not surprising that these patient cells are sensitive to UV radiation [111]. As with other diseases caused by defects in NER, these individuals also seem to be predisposed to sun-induced cancers [111]. Further studies suggested that the S228I pathogenic mutant was defective in NER due to disruption of PCNA interactions with DNA ligase 1, FEN1 and XP complementation group G (XPG) [111,185]. Ligase 1 and FEN1 also play an important role in Okazaki fragment processing. A combined defect in NER and Okazaki fragment processing may explain the broader array of symptoms seen in these patients as opposed to XP and CS patients. Conversely, symptoms of these patients seemed less severe than XP, CS or AT patients, suggesting that this *PCNA* mutant retains some DNA repair function.

### 2.8. RECQL4 Dysfunction Causes Three Overlapping Syndromes

Pathogenic variants in *RECQL4* have been linked to three syndromes: Rothmund–Thomson (RTS), Baller–Gerold (BGS) and RAPADILINO (RAdial hypo/aplasia, PAellae hypo/aplasia and cleft PAlate, DIarrhea and DIslocated joint, LIttle size and LImb malformation, NOse slender and NOrmal intelligence) (Figure 12 and Appendix A) [112,113,120,186,187,188,189]. All of these diseases are autosomal recessive and share growth retardation, cancer predisposition (osteosarcoma and lymphoma) and radial ray defects as symptoms [112,113,117,119,120,186,187,188,189]. RTS was the first to be described in the 19th century, but of course the gene affected in this disease was not found until more than a century later [120,187]. Kitao and coworkers investigated *RECQL4* because RTS patients have chromosomal instability, growth retardation, hypogonadism and cancer predisposition similar to individuals with Bloom or Werner syndrome [120]. Bloom and Werner helicases are members of the RECQ helicase family and RECQL4 was recently identified as a new member of this family [120]. In addition to the symptoms that RTS has in common with Bloom syndrome, Werner syndrome, BGS and RAPADILINO, RTS individuals can also have photosensitivity with poikiloderma, graying hair, alopecia, hyperpigmented patches, telangiectasias, dystrophic teeth and dystrophic nails [112,113,118,121,122,187,190,191,192,193]. Many unique genetic variants have been demonstrated to be causative for RTS [112,113,117,123,124,125,126,127,128,129,130,131,132,133,134,135,136,137,138,139,187,191]. Patients afflicted with RTS usually have compound heterozygous mutations in *RECQL4* with at least one nonsense mutation that causes a truncation, either deleting or disrupting the helicase domain (Figure 8 and Appendix A) [112,118,120].

The clinical features of BGS patients show significant overlap with both RAPADILINO and RTS, including radial ray defects, growth deficiency, gastrointestinal (GI) disturbances, anal anomalies, patellar abnormalities and poikiloderma [164]. The distinguishing feature of BGS is craniosynostosis, which can lead to death in early childhood without surgery [164]. Of note, many reported BGS cases are of terminated pregnancies, limiting the description of additional clinical findings [112,113,114,194]. As we increase our understanding of the genetic cause of BGS and RTS, it may be more appropriate to consider them as a single disease with variable expressivity. In fact, the similarities between BGS and RTS are so strong that, in 2015, Piard and colleagues suggested that BGS should be considered a severe form of RTS with craniosynostosis when caused by pathogenic variants of *RECQL4* [118]. As in RTS, the pathogenic variants are often inherited as compound heterozygous mutations and comprise missense, nonsense or splice site mutations (Figure 12 and Appendix A) [115,116,120,188]. It is important to note that, as with many diseases that were first described based on clinical characteristics, not all individuals with BGS and RTS have the same genetic cause. Ten percent of individuals clinically diagnosed with RTS have mutations in *anaphase promoting complex subunit 1* (*ANAPC1*) and 30% do not have a known genetic cause [195,196]. Similarly, patients with BGS carry mutations in *twist family BHLH transcription factor 1* (*TWIST1*), defects in which are known to cause craniosynostosis [170].

Whereas the mutation spectrum and patient population of RTS and BGS is quite broad, RAPADILINO occurs in a more homogenous Finnish population, with high prevalence of a splice site mutation in intron 7 of RECQL4 (Figure 8) [186]. This mutation results in the loss of exon 7 and an internal deletion of 44 amino acids within the N-terminal domain [186]. This region does not contain any known functional domains but was predicted to affect protein folding [186]. Subsequent studies demonstrated that the mutant is impaired for localization to the nucleus and sites of DNA damage [197,198]. Furthermore, in vitro studies with purified protein showed that the mutant lacked ATPase and ATP-dependent helicase activity [199]. RECQL4 can separate dsDNA when excess ssDNA is present with and without ATP, and surprisingly, the exon 7 mutant maintained this activity [190]. In fact, in assays without ATP, the mutant had higher activity than full-length RECQL4 [199]. Interestingly, some individuals with RAPADILINO are homozygous for the exon 7 skipping mutation, while others are compound heterozygous with a second pathogenic variant, most commonly a nonsense mutation (Figure 8 and Appendix A) [186]. A few individuals diagnosed with RAPADILINO do not carry this splice site mutation, but as poikiloderma, a distinguishing feature between RAPADILINO and RTS, does not typically appear in the first year of life, these individuals may be misdiagnosed [186,199]. Importantly, RAPADILINO patients do not have any skin or hair abnormalities as described in RTS [186,189].

Immunodeficiency has been reported in one RAPADILINO patient with decreases in T, B and NK cells [200]. Additionally, several case reports of RTS have been published reporting variable immune defects ranging from combined immune deficiency to immunoglobulin deficiency [201,202,203,204]. However, immune defects do not appear to be a consistent feature of either of these diseases. Both RTS and RAPADILINO are characterized by loss of RECQL4 helicase activity yet have discordant phenotypes. Recent studies suggested that the helicase domain is dispensable for cell viability and instead the N-terminus is required for efficient DNA replication as it has homology to *S. cerevisiae* Sld2, whereas the C-terminus is required for the DNA damage response [204,205]. Thus, it may be that the discordant phenotypes of RTS and RAPADILINO are due to separation-of-function mutations affecting either the N or C terminus of the protein.

## 3. Conclusions

Diseases caused by inherent defects in DNA replication genes exhibit a surprising diversity of symptoms. One overriding pathology seen in almost all of these diseases is growth restriction or short stature. This is consistent with the poor cell proliferation seen in many molecular studies of patient-derived cells and demonstrates the important role of efficient DNA replication in organismal growth. Other symptoms support the known functions of each protein defective in a disease. For instance, defects in pol ε, *PCNA* and *RECQL4* lead to increased risk for cancer. In Table 1 and Appendix A, we have compiled the symptoms reported for DNA replication-associated diseases. Not only can this serve as a diagnostic resource, but it highlights similarities and differences. In the future, individuals with phenotypes overlapping to those discussed here will be identified. When the genetic cause is discovered to reside in a poorly characterized gene such as *DONSON*, the understanding we have about the etiology of these diseases can be leveraged to predict the molecular function of a gene product. It remains unclear why variants in proteins with interconnected function cause distinct phenotypes and display an unexpectedly high degree of tissue specificity. Another remaining question in the field is how these pathogenic variants arose. Clearly, some of these diseases that occur in homogenous populations or large consanguineous families benefit from a founder effect. However, there are common mutations that repeatedly occur in unrelated patients, the most striking example of which is *POLD1* c.1686 + 32C > G in IMAGe syndrome. We suggest that it could be due to proofreading defects, failure of mismatch repair and/or residing in a mutagenesis hotspot.

An intriguing connection between MGS and ciliopathy is based on the observation that depletion of ORC1, ORC4, ORC6, CDT1 and CDC6 leads to reduced primary cilia formation [145]. The role of MCM proteins in cilia formation is also consistent with familial sensorineural hearing loss with MCM2 defects and adrenal disease in *MCM4* NKD patients. Defects in origin licensing factors result in skeletal, respiratory and genitourinary symptoms. Conversely, variants in origin firing factors and polymerases are strongly associated with immune defects. Interestingly, mutations in *RECQL4* have predominantly skeletal abnormalities rather than immunological effects, different from other factors involved in replication initiation. A similar picture emerges for the MCM2-7 complex. MCM4 and MCM5 are both part of the MCM2-7 double hexamer that is loaded onto DNA during origin licensing. As such, loss of these proteins would constitute an origin licensing defect. However, the *MCM4* mutation associated with NKD did not alter chromatin binding of MCM2-7 but rather impaired origin firing [81,82]. Thus, it appears that two categories exist: (1) abnormal replication origin licensing resulting in abnormal body patterning such as skeletal defects and (2) abnormal replication origin firing and elongation resulting in immunodeficiency. Surprisingly, molecular studies also indicate that defects in origin licensing do not necessarily result in genomic instability, whereas defects in origin firing and elongation do [81,85,93,98,120].

One possible explanation for the different tissue manifestations may be rooted in the patterns that different cell lineages utilize to assemble and activate origins of replication. As cells differentiate from a highly proliferative pluripotent stem cell, they alter replication timing in a cell-type-dependent manner as well as origin usage [206,207,208]. Furthermore, common fragile sites show lineage specificity, indicating that different regions of the genome are sensitive to perturbation of the replication program depending on the cell type [209]. It is therefore conceivable that differences in the replication program sensitize specific cell types to depletion of origin licensing or origin firing factors [209,210,211]. For example, we recently proposed that NK cells are sensitive to MCM10 depletion due to enhanced telomere erosion [85]. Hypothetically, a reduction in MCM10 and the resulting origin firing defects might disproportionally affect chromosome ends. This would diminish origin firing in regions critical for telomere replication causing the telomere-specific replication defects that we observed in MCM10-deficient model cell lines [85]. NK cells might be particularly sensitive to telomere erosion, as they already have shorter telomeres than other highly proliferative immune cells [174,175]. Thus, the loss of efficient origin firing and a short telomere “set point” might synergize to cause premature senescence during NK cell development.

While further molecular studies are needed to understand how defects in origin licensing and origin firing result in unique phenotypes, we propose that each cell type maintains a balance between origin licensing and origin firing to efficiently replicate their DNA. Furthermore, some cell types can withstand a reduction in origin licensing and/or origin firing due to their unique replication program. Disease arises when specific cell types cannot compensate for these replication defects, resulting in genomic instability, accumulation of replication stress and loss of progenitor cells. Exploring differences in origin licensing and efficiency of origin firing in different cell types would be informative for understanding both DNA replication-associated diseases and how lineage context affects the replication program.

## Figures and Tables

**Figure 1 ijms-22-00911-f001:**
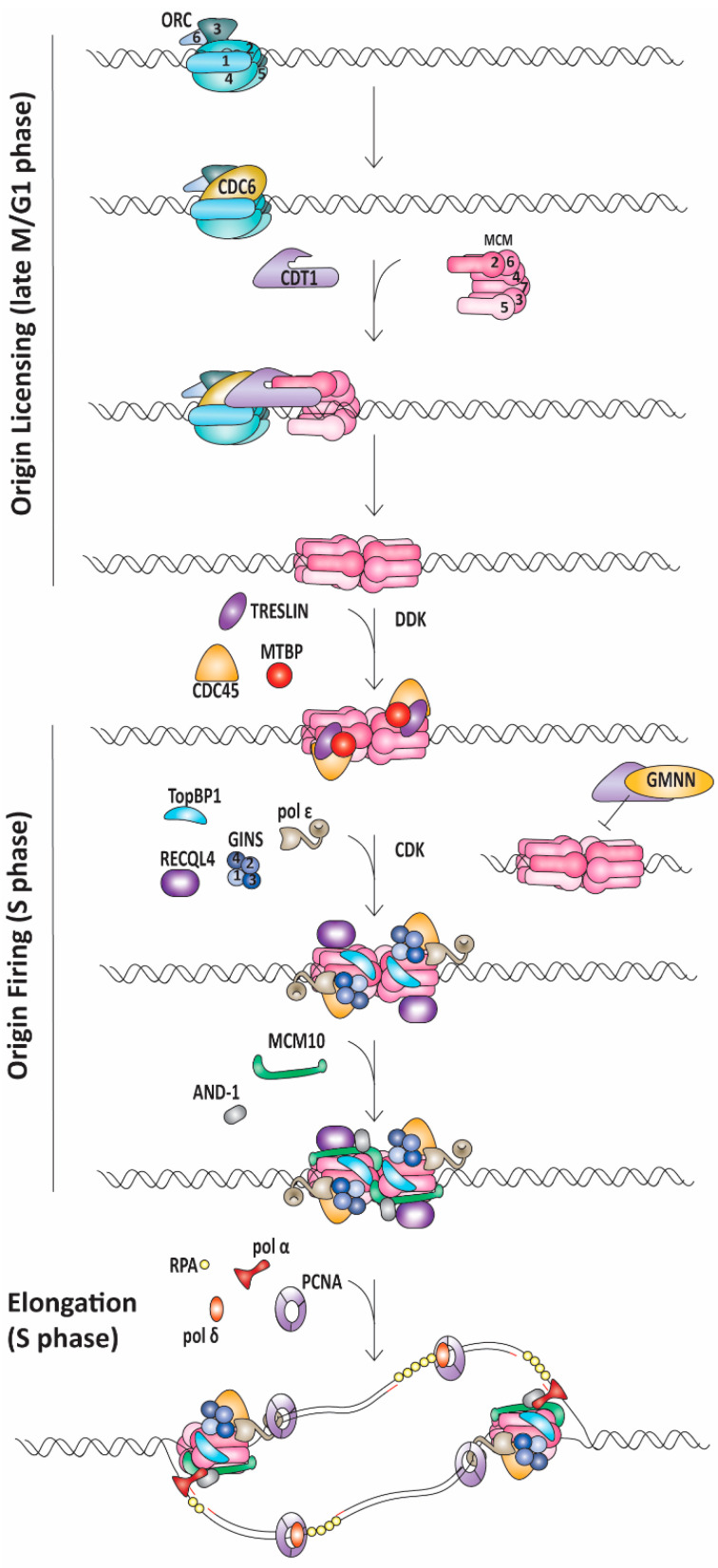
Origin licensing and firing require the coordinated action of multiple protein complexes. Origin licensing and firing require the coordinated action of multiple protein complexes. This cartoon depicts the steps and essential proteins involved in DNA replication origin licensing and firing, drawing from knowledge gained by studying yeast and human systems. Origin licensing occurs in the late M and G1 phase of the cell cycle when ORC1-6 binds to origins of replication. Together, CDC6 and ORC recruit CDT1 and MCM2-7, loading two MCM2-7 complexes onto dsDNA. In budding yeast, CDT1 binds to the MCM2-7 hexamer; however, these proteins have not been identified in a soluble complex without DNA in human cells. As cells transition into the S phase, Treslin, CDC45 and MTBP are recruited in a DDK-dependent manner. Pol ε and the GINS complex are recruited together with TopBP1 and RECQL4 in a CDK-dependent manner. During the S phase, licensing of additional origins is prevented by multiple mechanisms including sequestering of CDT1 by GMNN. Activation of DNA replication or origin firing requires MCM10, which aids in the bypass of the two CMG helicases past each other. MCM10 and AND-1 anchor pol α-primase to initiate DNA synthesis. As the CMG helicases progress in opposite directions, two replication forks form with pol α-primase, pol δ, pol ε and PCNA to promote DNA synthesis. Pol ε synthesizes the leading strand, while RPA binds single-stranded DNA on the lagging strand template until Okazaki fragments are produced by the consecutive action of pol α-primase and pol δ. Abbreviations: ORC, origin recognition complex; CDC, cell division cycle; CDT1, chromatin licensing and DNA replication factor 1; MCM, minichromosome maintenance; MTBP, Mdm2-binding protein; DDK, Dbf4-dependent kinase; GINS, go-ichi-ni-san; TopBP1, DNA topoisomerase II binding protein 1; RECQL4, ATP-dependent DNA helicase Q4; CDK, cyclin-dependent kinases; GMNN, geminin; AND-1, acidic nucleoplasmic DNA-binding protein 1; PCNA, proliferating cell nuclear antigen; RPA, replication protein A.

**Figure 2 ijms-22-00911-f002:**
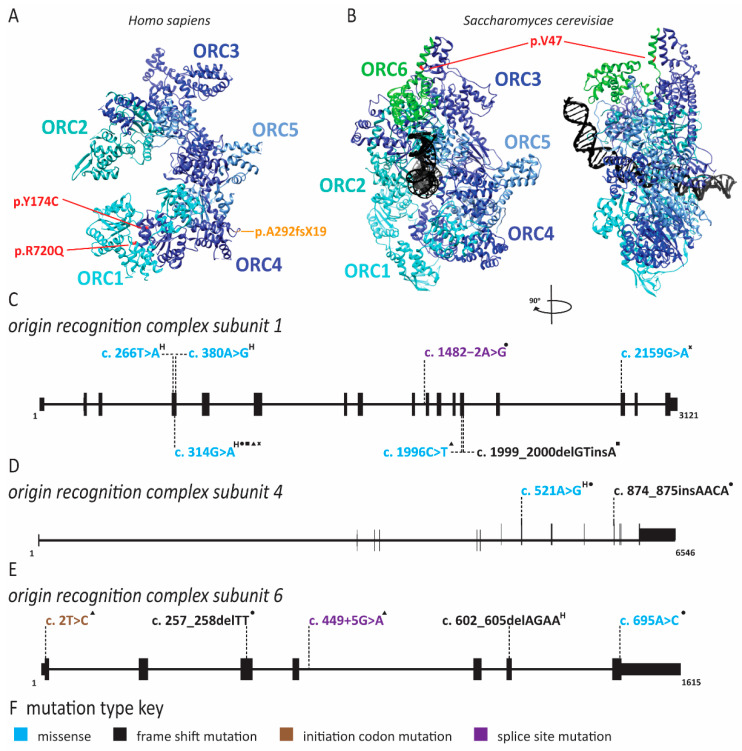
Mutations in *origin recognition complex* (*ORC*) subunits cause Meier-Gorlin syndrome. Pathogenic mutations in *ORC* subunits are depicted on a schematic of each gene and the crystal structure of either *Homo sapiens* or *Saccharomyces cerevisiae* proteins. Mutations that do not have corresponding amino acids in the structure or resulted in abnormal splicing are not depicted. For a complete list of mutations and associated protein changes, see Appendix A. Structures were generated with the Chimera program (http://www.cgl.ucsf.edu/chimera). Exon and intron schematics were generated with the UCSD genome browser. Reference sequences are ORC1 NM_004153.4, ORC4 NM_181742.4 and ORC6 NM_014321.4. (**A**) Crystal structure of human ORC subunits with mutations depicted, including missense mutations (red) and frame shift mutations (orange). The structure was generated using pdb file 5UJM. (**B**) Alignment of *Homo sapiens* and *Saccharomyces cerevisiae* ORC6 was completed using UniProt alignment tool (https://www.uniprot.org/align/). The amino acid corresponding to missense mutation p.Y232S (c.695A > C) is identified in red on the crystal structure of yeast ORC6. The structure was generated using pdb file 5zr1. Schematic of (**C**) *ORC1*, (**D**) *ORC4* and (**E**) *ORC6* genes depicted with exons as black boxes and introns as horizontal lines. Mutations are mapped to the appropriate gene regions. “H” superscript indicates a homozygous mutation. Compound heterozygous mutation pairs are indicated by superscript symbols. Each mutation combination is depicted, and certain alleles may be present in multiple combinations such as *ORC1* c.314G > A. The frequency of each allele in the affected population is not indicated. (**F**) For each gene, the color of the mutations indicates the type of mutation according to this key.

**Figure 3 ijms-22-00911-f003:**
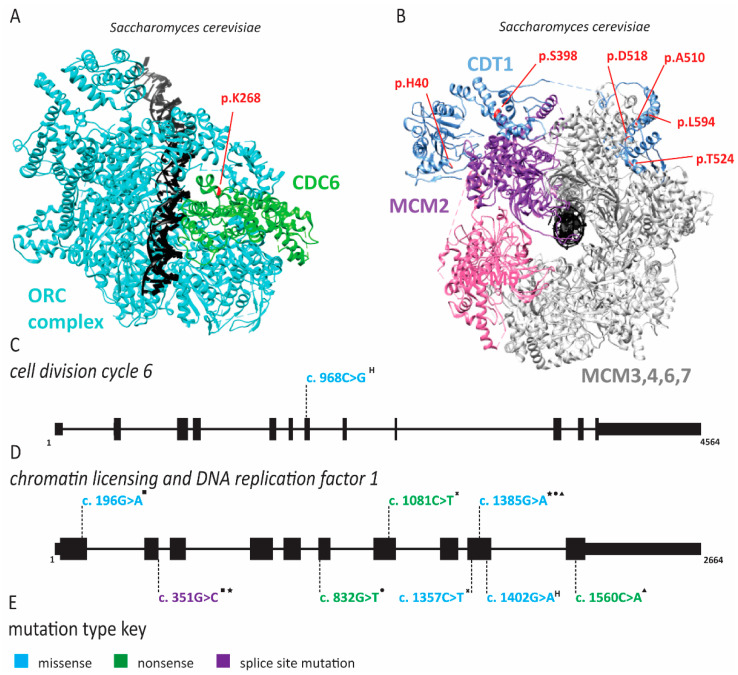
Mutations in *chromatin licensing and DNA replication factor 1* (*CDT1*) and *cell division cycle* 6 (*CDC6*) cause Meier-Gorlin syndrome. Pathogenic mutations of *CDC6* and *CDT1* are depicted on a schematic of each gene and crystal structure of *Saccharomyces cerevisiae* proteins. Mutations that do not have corresponding amino acids in the structure or resulted in abnormal splicing are not depicted. For a complete list of mutations and associated protein changes, see Appendix A. Alignment of *Homo sapiens* and *Saccharomyces cerevisiae* genes was completed using UniProt alignment tool (https://www.uniprot.org/align/). The structure was generated using pdb file 5v8f and the Chimera program (http://www.cgl.ucsf.edu/chimera). Exon and intron schematics were generated with the UCSD genome browser. Reference sequences are CDC6 NM_001254.4 and CDT1 NM_030928.4. (**A**). Crystal structure of yeast CDC6-ORC-MCM2-7-CDT1 complex (MCM2-7 and CDT1 not shown) with the amino acid associated with missense mutation p.T323R (c.968C > G) is depicted. (**B**) Crystal structure of yeast CDC6-ORC-MCM2-7-CDT1 complex (CDC6 and ORC not shown) with amino acids corresponding to missense and nonsense mutations in the human protein identified in red on the crystal structure of yeast CDT1. Schematic of (**C**) *CDC6* and (**D**) *CDT1* genes depicted with exons as black boxes and introns as horizontal lines. Mutations are mapped to the appropriate gene regions. “H” superscript indicates a homozygous mutation. Compound heterozygous mutation pairs are indicated by superscript symbols. Each mutation combination is depicted, and certain alleles may be present in multiple combinations. The frequency of each allele in the affected population is not indicated. (**E**) For each gene, the color of the mutations indicates the type of mutation according to this key.

**Figure 4 ijms-22-00911-f004:**
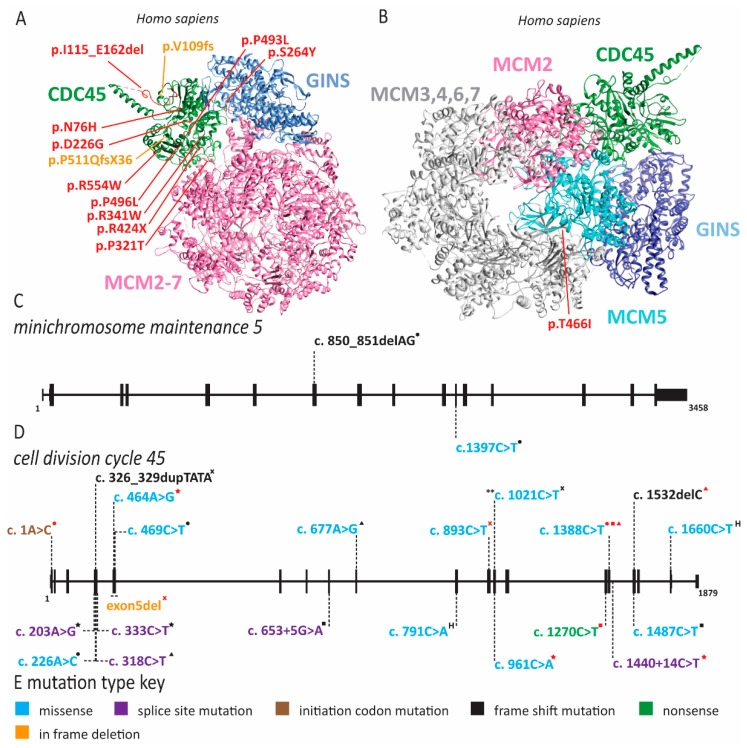
Mutations in helicase component *minichromosome maintenance 5* (*MCM5*) and *cell division cycle 45* (*CDC45*) cause Meier-Gorlin syndrome. Pathogenic mutations of *MCM5* and *CDC45* are depicted on a schematic of each gene and the crystal structure of *Homo sapiens* proteins. Mutations that do not have corresponding amino acids in the structure or resulted in abnormal splicing are not depicted. For a complete list of mutations and associated protein changes, see Appendix A. The structure was generated using pdb file 6xtx and the Chimera program (http://www.cgl.ucsf.edu/chimera). Exon and intron schematics were generated with the UCSD genome browser. Reference sequences are CDC45 NM_003504.4 and MCM5 NM_006739.4. (**A**) Crystal structure of human CDC45 with mutations depicted including missense mutations (red) and frame shift mutations (orange). Note p.115_E162del is a large deletion of exon 5, with all lost amino acids highlighted in red. (**B**) Crystal structure of human MCM5 with missense mutation p.T466I (c.1397C > T) depicted in red. Schematic of (**C**) *MCM5* and (**D**) *CDC45* genes depicted with exons as black boxes and introns as horizontal lines. Mutations are mapped to the appropriate gene regions. “H” superscript indicates a homozygous mutation. ** denotes different than reported due to different reference sequence. Compound heterozygous mutation pairs are indicated by superscript symbols. Each mutation combination is depicted, and certain alleles may be present in multiple combinations. The frequency of each allele in the affected population is not indicated. (**E**) For each gene, the color of the mutations indicates the type of mutation according to this key.

**Figure 5 ijms-22-00911-f005:**
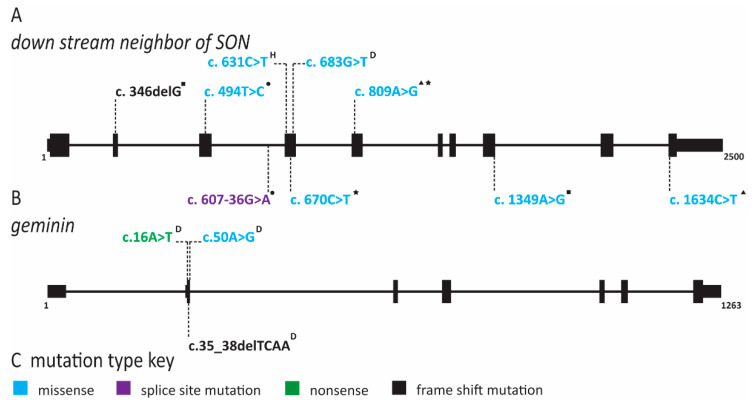
Mutations in *downstream neighbor of son* (*DONSON*) and *geminin* (*GMNN*) cause Meier-Gorlin syndrome. Exon and intron schematic was generated with the UCSD genome browser. Reference sequences are DONSON NM_017613.4 and GMNN NM_015895.5. Schematic of (**A**) *DONSON* and (**B**) *GMNN* genes depicted with exons as black boxes and introns as horizontal lines. Mutations are mapped to the appropriate gene regions. “H” superscript indicates a homozygous mutation. “D” superscript indicates a heterozygous dominant mutation. Compound heterozygous mutation pairs are indicated by superscript symbols. Each mutation combination is depicted, and certain alleles may be present in multiple combinations. The frequency of each allele in the affected population is not indicated. (**C**) For each gene, the color of the mutations indicates the type of mutation according to this key.

**Figure 6 ijms-22-00911-f006:**
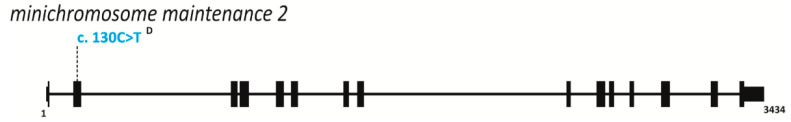
*Minichromosome maintenance* 2 heterozygous variant is linked to familial deafness. Exon and intron schematic was generated with the UCSD genome browser. Reference sequence is MCM2 NM_004526.4. Schematic of *MCM2* gene is depicted with exons as black boxes and introns as horizontal lines. Missense mutation is mapped to the appropriate gene regions. “D” superscript indicates a heterozygous dominant mutation.

**Figure 7 ijms-22-00911-f007:**
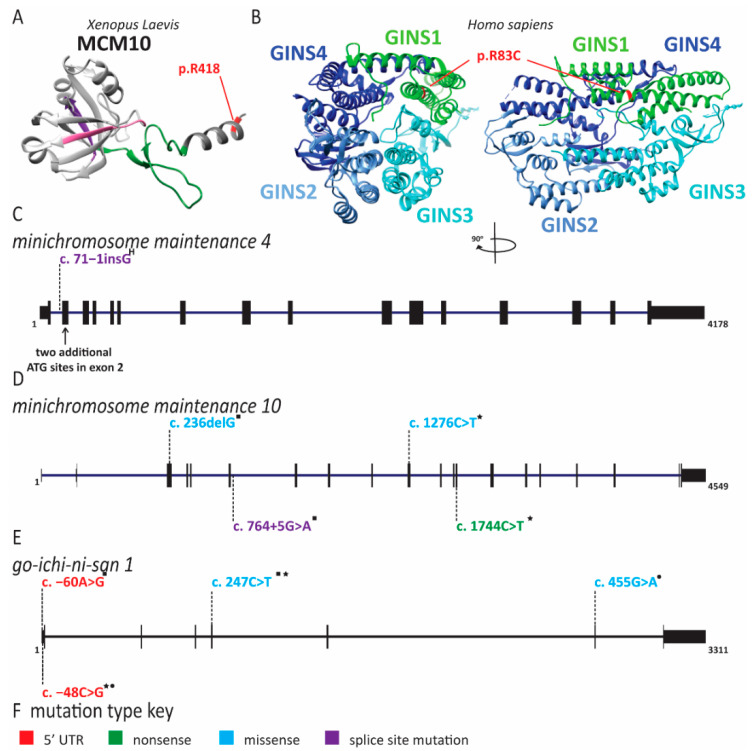
Natural killer cell deficiency is caused by pathogenic variants disrupting DNA replication initiation. Pathogenic mutations in replication initiation factors are depicted on a schematic of each gene and the crystal structure of either *Homo sapiens* or *Xenopus laevis* proteins. Mutations that do not have corresponding amino acids in the structure or resulted in abnormal splicing are not depicted. For a complete list of mutations and associated protein changes, see Appendix A. Structures were generated with the Chimera program (http://www.cgl.ucsf.edu/chimera). Exon and intron schematics were generated with the UCSD genome browser. Reference sequences are MCM4 NM_005914.4, MCM10 NM_018518.5 and GINS1 NM_021067.5. (**A**) Alignment of *Homo sapiens* and *Xenopus laevis* MCM10 was completed using UniProt alignment tool (https://www.uniprot.org/align/). Crystal structure of the internal domain of *Xenopus laevis* MCM10 is shown with PCNA binding peptide box in pink, heat shock protein 10 like domain in purple, and the first zinc finger in green. The *Xenopus laevis* amino acid corresponding to missense mutation p.R426C (c.1276C > T) is depicted in red. The structure was generated using pdb file 3ebe. (**B**) Missense mutation p.R83C (c.247C > T) in GINS1 is identified in red on the crystal structure of GINS complex. The structure was generated using pdb file 2q9q. Schematic of (**C***) MCM4*, (**D**) *MCM10* and (**E**) *GINS1* genes depicted with exons as black boxes and introns as horizontal lines. Mutations are mapped to the appropriate gene regions. For *MCM4*, two additional ATG sites that result in shorter isoforms occur in exon 2. “H” superscript indicates a homozygous mutation. Compound heterozygous mutation pairs are indicated by superscript symbols. Each mutation combination is depicted, and certain alleles may be present in multiple combinations. The frequency of each allele in the affected population is not indicated. (**F**) For each gene, the color of the mutations indicates the type of mutation according to this key.

**Figure 8 ijms-22-00911-f008:**
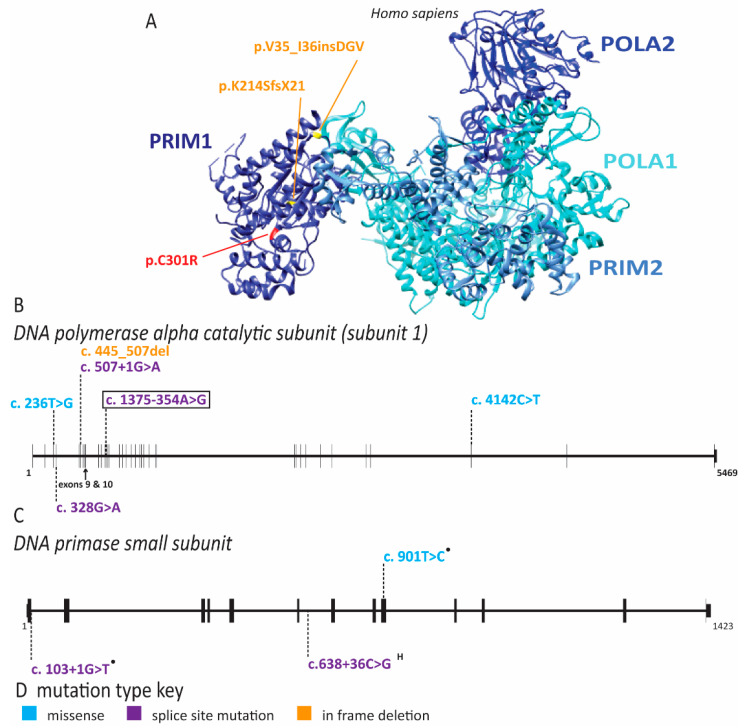
Pathogenic variants in pol α cause a variety of diseases with high prevalence of immunodeficiency. Pathogenic mutations in pol α subunits are depicted on a schematic of each gene and the crystal structure *of Homo sapiens* proteins. Mutations that do not have corresponding amino acids in the structure or resulted in abnormal splicing are not depicted. For a complete list of mutations and associated protein changes, see Appendix A. Structures were generated with the Chimera program (http://www.cgl.ucsf.edu/chimera). Exon and intron schematics were generated with the UCSD genome browser. Reference sequences are POLA1 NM_016937.4 and PRIM1 NM_000946.3. (**A**) Crystal structure of human pol α subunits with mutations depicted including missense mutations (red) and frame shift mutations (orange). The structure was generated using pdb file 5exr. Schematic of (**B**) POLA1 and (**C**) PRIM1 genes depicted with exons as black boxes and introns as horizontal lines. Mutations are mapped to the appropriate gene regions. Note that *POLA1* is on the X chromosome and has X-linked inheritance. Boxed alleles on *POLA1* schematic occur in individuals with XLPRD while the remaining mutations occur in individuals with Van Esch-O’Driscoll disease. Exons 9 and 10 of *POLA1* appear as a single line in this schematic. “H” superscript indicates a homozygous mutation. Compound heterozygous mutation pairs are indicated by superscript symbols. Each mutation combination is depicted, and certain alleles may be present in multiple combinations. The frequency of each allele in the affected population is not indicated. (**D**) For each gene, the color of the mutations indicates the type of mutation according to this key.

**Figure 9 ijms-22-00911-f009:**
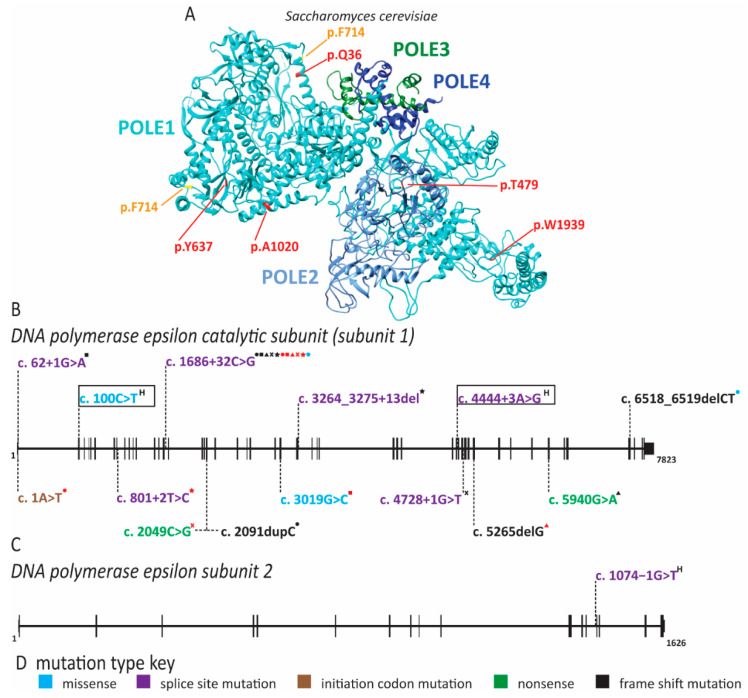
Pathogenic variants in pol ε cause immunodeficiency. Pathogenic mutations in pol ε subunits are depicted on a schematic of each gene and the crystal structure of *Saccharomyces cerevisiae* proteins. Mutations that do not have corresponding amino acids in the structure or resulted in abnormal splicing are not depicted. For a complete list of mutations and associated protein changes, see Appendix A. Structures were generated with the Chimera program (http://www.cgl.ucsf.edu/chimera). Exon and intron schematics were generated with the UCSD genome browser. Reference sequences are POLE1 NM_006231.4 and POLE2 NM_001197331.3. (**A**) Alignment of *Homo sapiens* and *Saccharomyces cerevisiae* POLE1 and POLE2 was completed using UniProt alignment tool (https://www.uniprot.org/align/). Crystal structure of yeast pol ε subunits with amino acids corresponding to missense (red), nonsense mutations (red) and frame shift mutations (orange) depicted. The structure was generated using pdb file 6wjv. Schematic of (**B**) *POLE1* and (**C**) *POLE2* genes depicted with exons as black boxes and introns as horizontal lines. Mutations are mapped to the appropriate gene regions. Boxed alleles on *POLE1* occur in individuals with FILS syndrome. “H” superscript indicates a homozygous mutation. Compound heterozygous mutation pairs are indicated by superscript symbols. Each mutation combination is depicted, and certain alleles may be present in multiple combinations such as c.1686 + 32C > G which is shared by all individuals with IMAGe syndrome. The frequency of each allele in the affected population is not indicated. (**D**) For each gene, the color of the mutations indicates the type of mutation according to this key.

**Figure 10 ijms-22-00911-f010:**
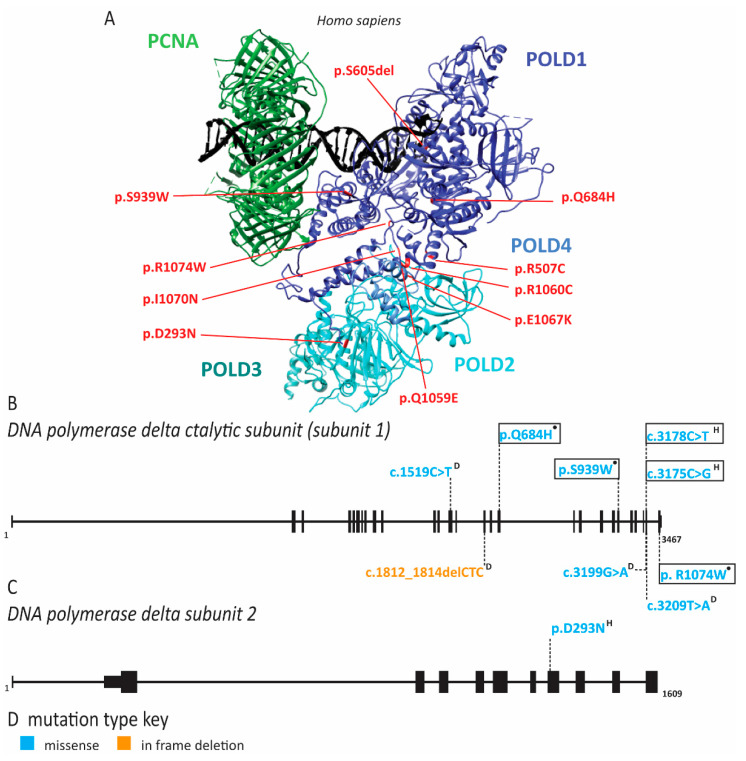
Pathogenic variants in pol δ subunits cause human disease. Pathogenic mutations in pol δ subunits are depicted on a schematic of each gene and the crystal structure of *Homo sapiens* pol δ complex. Mutations that do not have corresponding amino acids in the structure or resulted in abnormal splicing are not depicted. For a complete list of mutations and associated protein changes, see Appendix A. Structures were generated with the Chimera program (http://www.cgl.ucsf.edu/chimera). Exon and intron schematics were generated with the UCSD genome browser. Reference sequences are POLD1 NM_001256849.1 and POLD2 NM_006230.4. (**A**) Crystal structure of human pol δ subunits with mutations depicted including missense and nonsense mutations depicted in red and frame shift mutations depicted in orange. The structure was generated using pdb file 6s1m. Schematic of (**B**) *POLD1* and (**C**) *POLD2* genes depicted with exons as black boxes and introns as horizontal lines. Mutations are mapped to the appropriate gene regions. Boxed alleles on *POLD1* occur in individuals with immune deficiency. Note that the black circle symbol indicates mutations found together in an individual, p.Q684H and p.S939W were found in cis while p.R1074W was found in trans. Compound heterozygous mutation pairs are indicated by superscript symbols. Each mutation combination is depicted, and certain alleles may be present in multiple combinations. The frequency of each allele in the affected population is not indicated. (**D**) For each gene, the color of the mutations indicates the type of mutation according to this key.

**Figure 11 ijms-22-00911-f011:**
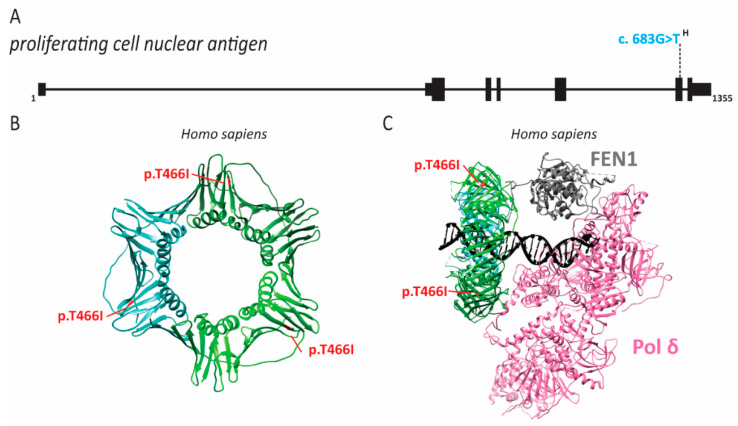
*PCNA* pathogenic variant is on the outer surface of PCNA. Pathogenic mutation in PCNA is depicted on a schematic of the gene and crystal structure of *Homo sapiens* proteins. Structures were generated with the Chimera program (http://www.cgl.ucsf.edu/chimera). Exon and intron schematics were generated with the UCSD genome browser. Reference sequence is NM_002592.2. Schematic of the (**A**) *PCNA* gene is depicted with exons as black boxes and introns as horizontal lines. Missense mutation is mapped to the appropriate gene regions. “H” superscript indicates a homozygous mutation. (**B**) Crystal structure of human PCNA with missense mutation depicted in red. The structure was generated using pdb file 6fcm. (**C**) Crystal structure of human PCNA in complex with FEN1 and pol δ with missense mutation depicted in red. The structure was generated using pdb file 6tnz.

**Figure 12 ijms-22-00911-f012:**
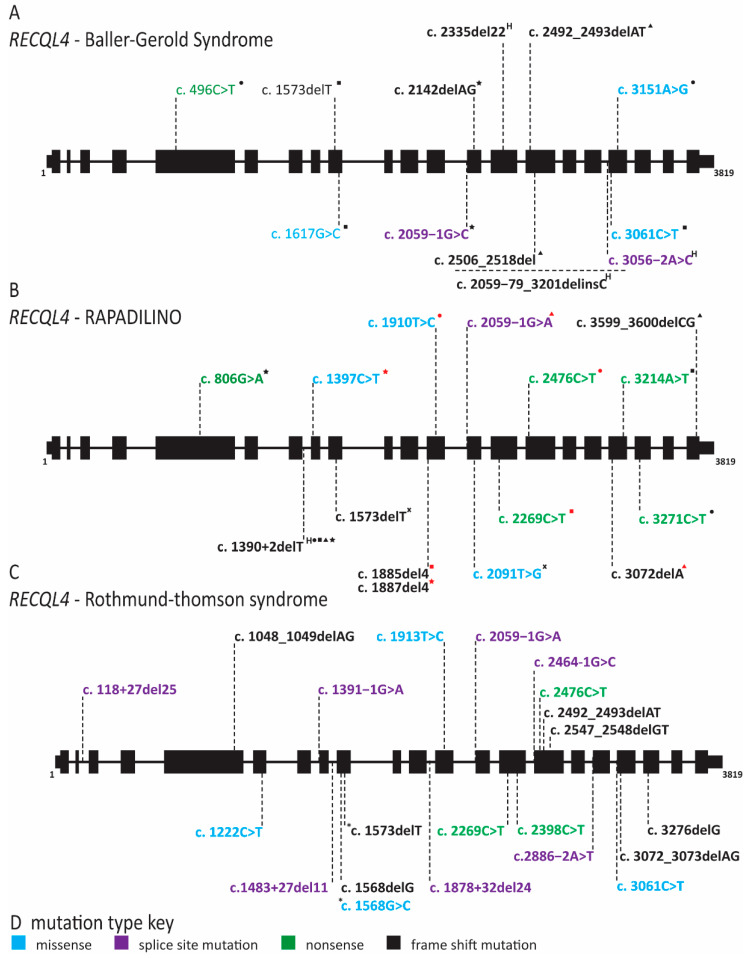
Pathogenic variants in *RECQL4* are associated with three different syndromes, Rothmund–Thomson (RTS), Baller–Gerold (BGS) and RAPADILINO (RAdial hy-po/aplasia, PAellae hypo/aplasia and cleft PAlate, DIarrhea and DIslocated joint, LIttle size and LImb malformation, NOse slender and NOrmal intelligence). Mutations of *RECQL4* in BGS, RAPADILINO and RTS are depicted on a schematic of the gene and crystal structure of *Homo sapiens* RECQL4. For a complete list of mutations and associated protein changes, see Appendix A. The structure was generated using pdb file 5lst and the Chimera program (http://www.cgl.ucsf.edu/chimera). Exon and intron schematics were generated with the UCSD genome browser. Reference sequence is RECQL4 NM_004260.4. Schematic of *RECQL4* gene for (**A**) BGS and (**B**) RAPADILINO is depicted with exons as black boxes and introns as horizontal lines. Mutations are mapped to the appropriate gene regions. “H” superscript indicates a homozygous mutation. Compound heterozygous mutation pairs are indicated by superscript symbols. Each mutation combination is depicted, and certain alleles may be present in multiple combinations. The frequency of each allele in the affected population is not indicated. (**C**) Mutations that occur in multiple individuals with RTS are depicted. “*” indicates that these mutations occur in cis, c.1573delT does occur alone in some individuals (**D**) For each gene, the color of the mutations indicates the type of mutation according to this key. Crystal structure of human RECQL4 with the helicase domain highlighted in blue, missense and nonsense mutations highlighted in red and frameshift mutations highlighted in orange for (**E**) BGS, (**F**) RAPADILINO and (**G**) RTS. For RTS, one amino acid indicated in orange on the ribbon structure has two mutations associated with it. Mutations that do not have corresponding amino acids in the structure or resulted in abnormal splicing are not depicted.

**Table 1 ijms-22-00911-t001:** Common symptoms occurring in diseases caused by defects in DNA replication. This table compiles common symptoms of diseases discussed in this review. A more exhaustive list of symptoms can be found in Appendix A.

	Gene(s) Affected	MIM ID	*Inheritance*	IUGR	Short Stature	Microcephaly	Craniosynostosis	Facial Features	Skin Changes	Radial Ray Defects	A/Hypoplastic Patella	Adrenal Insufficiency	GU Abnormalities (male)	ID	DD	Hearing Loss	Immune Defects	Anorectal Malformation	Feeding Difficulties	Diarrhea	Premature Aging	Lipodystrophy	Cancer Predisposition	Other symptoms	References
MGS	ORC1	224690	*AR*	x	x	x		microtia microstomia full lipsret rognathia micrognathia			x		x		x				x						[67,68,72,73,74]
	ORC4	613800	*AR*	x	x						x				x				x						[67,68,72]
	ORC6	613803	*AR*	x	x						x		x		x				x						[67,72,75]
	CDT1	613804	*AR*	x	x						x		x		x				x						[67,68,72,76]
	CDC6	613805	*AR*	x	x						x		x						x						[72,76]
	MCM5	617564	*AR*	x	x						x		x												[70]
	CDC45	617063	*AR*	x	x	x	x				x		x					x	x						[69,77]
	GMNN	616835	*AD*	x	x						x		x	x					x						[71]
	DONSON		*AR*	x	x	x					x				x	x									[76,78]
NKD	MCM4	609981	*AR*	x	x	x		large forehead, thin upper lip				x		x	x		x						+/−		[79,80,81,82]
	GINS1	617827	*AR*	x	x			flat nasal bridge, round nose tip, blepharophimosis, posteriorly rotated ears, thin upper lip	E								x				+/−				[83]
	MCM10		*AR*	UR	UR			none noted									x								[84,85]
XLPRD	POLA1	301220	*XLR*					Blepharophimosis, flared eyebrows, upswept hair									x			x				sterile multiorgan inflammation	[86,87,88,89,90]
VEODS	POLA1	301030	*XLR*		x	x		inconsistent						x	x		+/−								[91,92]
	PRIM1		*AR*		x	x		prominent forehead, triangular face, blepharophimosis, micrognathia, small-low set ears					x		x		x		x			x			[93]
FILS	POLE1	615139	*AR*		x			malar hypoplasia high forehead	L								x								[94,95,96]
IMAGe	POLE1	618336	*AR*	x	x	x		Micrognathia, long thin nose, short wide neck, small low-set, posteriorly rotated ears		other		x	x				x		x				+/−		[97]
	POLE2		*AR*		x			low anterior hairline, flat supraorbital ridges, downturned corners of mouth, short philtrum									x								[98]
MDP	POLD1	615381	*AD*		x			mandibular hypoplasia, pinched nose, microstomia, bird like facies, prominent eyes	T				x			x					x	x			[99,100,101,102,103,104,105,106,107]
	POLD1		*AR*		x			none noted						x		x	x								[108,109,110]
	POLD2		*AR*		x	x		none noted						x		x	x								[108]
	PCNA	615919	*AR*		x			none noted	T						x	x					x		x	neurodegeneration, ataxia	[111]
BGS	RECQL4	218600	*AR*	x	x		x	present but variable	P	x					x	x		x	x				x		[112,113,114,115,116]
RAPA	RECQL4	266280	*AR*	x	x			long face, long slender nose, narrow palpebral fissures, arched palate		x	x						+/−		x	x			x		[112,117,118,119]
RTS	RECQL4	268400	*AR*	x	x	x		present but variable	T, P	x	x					x	+/−	x	x	x			x	alopecia	[112,114,117,118,119,120,121,122,123,124,125,126,127,128,129,130,131,132,133,134,135,136,137,138,139,140,141]
	MCM2	616968	*AD*					none noted								x									[142]

MIM ID, Mendelian Inheritance in Man identification number; IUGR, intrauterine growth restriction; ID, intellectual disability; DD, developmental delay; VEODS, Van Esch–O’Driscoll syndrome; RAPA, RAPADILINO; MGS, Meier-Gorlin; NKD, natural killer cell deficiency; XLPRD, X-linked pigmentary reticulate disorder; FILS, facial dysmorphism, immunodeficiency, livedo and short stature; IMAGe, intrauterine growth restriction, metaphyseal dysplasia, adrenal hypoplasia congenita and genitourinary abnormalities syndrome; MDP, mandibular hypoplasia, deafness, progeroid features; BGS, Baller–Gerold; RTS, Rothmund–Thomson;AR, autosomal recessive; AD, autosomal dominant; XLR, X-linked recessive; UR, unreported; E, eczema; L, livedo; T, telangiectasia; P, poikiloderma.

## Data Availability

Not applicable.

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
