# Peer review of "Congenital Diseases of DNA Replication: Clinical Phenotypes and Molecular Mechanisms"

_ijms, 2021, doi:10.3390/ijms22020911_

Round 1

Reviewer 1 Report

  • Originality/Novelty: This review provides a good summary of the current state of the field.
  • Significance: The topics and studies highlighted in this review are of high significance, and therefore this review is timely.
  • Quality of Presentation: Presentation quality is good and figures are of high quality.
  • Scientific Soundness: The studies and data be referenced in this review are scientifically sound.
  • Interest to the Readers: This review will be of interest to readers from multiple disciplines.
  • Overall Merit: There is overall benefit to publishing this work, as it highlights recent results and outstanding questions that span multiple disciplines.

Brief summary:

Schmit and Bielinsky provide an extremely comprehensive summary of what is currently known about the connection between mutation of various DNA replication proteins and congenital diseases, including clinical descriptions and mechanistic insight into specific aspects of DNA replication that are known to be perturbed. They provide clear visual representation of the locations of a number of these mutations on X-ray crystallographic structures solved for the human protein or other eukaryotic homologs. This review provides an excellent overview of the current understanding of how replication initiation or elongation can be affected by mutation of critical replication factors.

Specific comments:

This review by Schmit and Bielinsky does an outstanding job of summarizing the important issues related to how defects in DNA replication can lead to disease. Clinical phenotypes are discussed, and descriptions of the molecular mechanisms by which different mutations disrupt essential replication proteins and pathways are described in detail. The amount of information included and the level of detail seems appropriate. Reference citations are satisfactory. This review will be of broad interest to clinicians, geneticists, and molecular biologists.

While this review provides a very thorough description of the clinical phenotypes presented in the diseases described, there are sections where a discussion of the molecular mechanisms contributing to the disease could be further developed (when information is available). For example, the authors describe how the mutation affects gene/protein expression, stability and splicing, but there are times where effects on protein function (enzymatic activity) or protein-protein interactions might be relevant. Wherever possible, a discussion of data that relates to how the specific mutations affect protein function could be extremely informative to the biochemists and structural biologists reading the review. The sections on DNA polymerase delta and RECQL4 both provide through descriptions of effects on protein function, and it would be great if this could be extended to other proteins where such information has been published.

Replication initiation, elongation and termination are all nicely discussed in the introduction, however diseases related to defects in only the first two processes described in the review. Have any diseases associated with replication termination been identified?

One general question is whether the authors can provide any information on what is known regarding the sources of the mutations identified in these replication genes. Are they thought to be associated with defective DNA polymerization, proofreading or DNA mismatch repair?

It is difficult to get a feel for how frequent these mutations occur in the general population – this information would be informative. As would some discussion of geographic diversity.

Author Response

Response to reviewer #1

We would like to thank the reviewer for the thoughtful comments and feedback. Please note that line numbers refer to the copy of the manuscript with tracked changes.

Reviewer comments are italicized and in blue.

“While this review provides a very thorough description of the clinical phenotypes presented in the diseases described, there are sections where a discussion of the molecular mechanisms contributing to the disease could be further developed (when information is available). For example, the authors describe how the mutation affects gene/protein expression, stability and splicing, but there are times where effects on protein function (enzymatic activity) or protein-protein interactions might be relevant. Wherever possible, a discussion of data that relates to how the specific mutations affect protein function could be extremely informative to the biochemists and structural biologists reading the review. The sections on DNA polymerase delta and RECQL4 both provide through descriptions of effects on protein function, and it would be great if this could be extended to other proteins where such information has been published.”

We appreciate the reviewer’s desire for more discussion of molecular mechanisms of diseases. In our original manuscript we had included information on defects in protein function based on published reports of either a single or several mutations suggestive of a common defect. We have added additional discussion on ORC1 and ORC6 mutations in MGS and their effects on the function of these proteins. Please see lines 369-377.

“Replication initiation, elongation and termination are all nicely discussed in the introduction, however diseases related to defects in only the first two processes described in the review. Have any diseases associated with replication termination been identified?”

We have added the following statement for clarification: “Currently, there are no known congenital diseases caused by disruption of replication termination.” Lines 213-215.

“One general question is whether the authors can provide any information on what is known regarding the sources of the mutations identified in these replication genes. Are they thought to be associated with defective DNA polymerization, proofreading or DNA mismatch repair?”

We appreciate the author’s interest in how the various mutations arise as well as the prevalence of these mutations in the general population. Unfortunately, little is known about how these mutations first occur, however in recognition of this unanswered and interesting question, we have included it in our conclusion section lines 1106-1113.

“It is difficult to get a feel for how frequent these mutations occur in the general population – this information would be informative. As would some discussion of geographic diversity.”

Some information is reported on the mutation frequency for various diseases, and we have included this where available in Supplemental Table 1.

Reviewer 2 Report

Schmit and Bielinsky, Congenital diseases of DNA replication: clinical phenotypes and molecular mechanisms

In this review manuscript, the authors provided an in-depth description of disease phenotypes caused by genetic mutations in factors involved in the process of DNA replication. The authors started with an overview of the known mechanisms of eukaryotic DNA replication initiation, elongation, and termination in section 1 (“Introduction” section). Then, in section 2 (“Diseases” section), the authors introduced various genetic diseases with mutations in factors involved in DNA replication regulation. These diseases exhibit varying degrees of severity, with many of them affecting only specific tissues even though the mutations should affect the process of DNA replication in all tissues. Lastly, in section 3 (“Conclusions” section; by the way, it says section 5 on line 1054 and should be corrected), the authors re-emphasized that the phenotypes are surprisingly different among different mutations and speculated that the varying phenotypes could be a consequence of the difference in replication timing programs, which are cell-type specific.

Overall, I think the manuscript comprehensively covers various genetic diseases related to DNA replication and will be useful to a broad readership, from researchers to clinicians. However, while the article is informative, it is certainly lengthy (45 pages!). It is also not very insightful either. For instance, it wasn’t clear to me why factor-specific and cell-type specific phenotypes may be the consequence of cell-type specific DNA replication timing programs. Some additional things need to be fixed. I suggest the following to improve the manuscript.

Major points:

1. Lack of insight

While the article is rich in information, it wasn’t clear to me why factor-specific and cell-type specific phenotypes may be the result of cell-type specific DNA replication timing programs (line 1102–1104), which was rather abruptly brought up on the last page. The authors should be more specific and provide a further explanation (in the Conclusion and Introduction sections). On a related note, it was not clear to me why the paragraph also discusses Reference 111 on MCM10-depleted NK cells. How is ref. 111 related to the DNA replication timing discussion?

I also felt that the discussion on the balance between origin licensing and origin firing to efficiently replicate genomic DNA and the cell-type specific nature of this balance is not well thought out. If they were to speculate on this, they would have to explain a bit more about what is currently known, perhaps in the Introduction section.

Without these corrections, the article remains a collection of facts (which is nice) but without much insight, contrary to what the authors have tried to do.

2. Are the phenotypes really factor-specific and cell-type specific?

I suppose they are, but to what extent? For instance, immune cells seem to be unaffected in MGS according to the authors’ literature survey, but I wonder if any research articles are showing that they are really unaffected? I mean, in Table 1, there are only “X” and “+/-“ but no “N/A”. Do the blanks all mean those phenotypes were unaffected? Alternatively, if some of them have never been tested, the table could be very misleading.

3. Figure 1 and the model organism

When explaining the mechanism of DNA replication, the authors should specify the model organism (in the figure title/legend as well as in the main text) so as not to confuse the readers. Figure 1 basically describes our current knowledge of DNA replication initiation mechanism based largely on studies carried out in budding yeast but it shows human proteins. Nonetheless, at times the authors talk as if this figure is describing the budding yeast replication model and use the yeast gene nomenclature (line 115 paragraph, for instance). They provide some explanation in a paragraph that starts on line 150 but it seems a bit late and out of place.

4. Termination

Replication termination is poorly discussed – only one short paragraph (line 189). For instance, they never talk about MiDAS. 

Line 510–513: this seems not true. Two recent back-to-back Cell Research articles in 2020 have addressed this.

5. Lengthiness

- I felt that some parts of this review article focused too much on very basic textbook knowledge and history of a particular biological phenomenon and may not be totally necessary. At least, they could be shortened.

- There are unnecessary blank spaces here and there (p. 13, 17, 19, 20, 23, 26, 30, and on every page on the left side), which could be removed. These blanks also make it very difficult to read the manuscript. 

- There are too many figures (12 figures) but I did not find Figs. 2–11 to be particularly informative. One could read the piece without going through the figures. Perhaps most of them could become Supplementary Figures.

Other points:

1. Page numbers are messed up after page 17 (around line 630).

2. The justification should be consistent throughout the manuscript. Currently, the text is full-justified except for lines 1055–1073, while the figure legends are left-justified. 

3. Line 124: ishi > ichi

4. Line 359: CDC6 primary cilia > CDC6, primary cilia (comma)

5. Lines 668, 838: unnecessary space

6. References 16 and 17 are duplicated

Author Response

Response to reviewer #2.

We would like to thank the reviewer for taking the time to provide detailed feedback. Please note that line numbers refer to the copy of the manuscript with tracked changes.

  1. Lack of insight

While the article is rich in information, it wasn’t clear to me why factor-specific and cell-type specific phenotypes may be the result of cell-type specific DNA replication timing programs (line 1102–1104), which was rather abruptly brought up on the last page. The authors should be more specific and provide a further explanation (in the Conclusion and Introduction sections). On a related note, it was not clear to me why the paragraph also discusses Reference 111 on MCM10-depleted NK cells. How is ref. 111 related to the DNA replication timing discussion?

 I also felt that the discussion on the balance between origin licensing and origin firing to efficiently replicate genomic DNA and the cell-type specific nature of this balance is not well thought out. If they were to speculate on this, they would have to explain a bit more about what is currently known, perhaps in the Introduction section.

 Without these corrections, the article remains a collection of facts (which is nice) but without much insight, contrary to what the authors have tried to do.

We regret that this reviewer found our manuscript to lack insight. In writing this review we aimed to provide a comprehensive discussion of diseases caused by defects in DNA replication to a broad audience, including physicians who may not have a significant background in the molecular aspects of DNA replication and basic research scientist who may not be aware of the clinical phenotypes. Additionally, our conclusions highlight intriguing and unanswered questions for the field. To increase clarity of this section, we have edited it to deepen the discussion on how altered replication dynamics may result in the cell type specific effects seen in the clinical presentations (lines 1136-1156). We have clarified in line 1145 that the discussion of MCM10 depleted NK cells is meant to be an example for the cell-type specific manifestation. Reference 111 (now 113) is related to our work on telomere length in MCM10 deficient cells.

  1. Are the phenotypes really factor-specific and cell-type specific?

I suppose they are, but to what extent? For instance, immune cells seem to be unaffected in MGS according to the authors’ literature survey, but I wonder if any research articles are showing that they are really unaffected? I mean, in Table 1, there are only “X” and “+/-“ but no “N/A”. Do the blanks all mean those phenotypes were unaffected? Alternatively, if some of them have never been tested, the table could be very misleading.

The reviewer questions if the phenotypes observed are truly factor and cell type specific and suggests Table 1 is misleading. As discussed in the legend to Table 1, this table addresses the common symptoms associated with each disease and is not exhaustive of every patient (the comprehensive data is presented in Supplemental Table 2) and thus a blank space indicates that a symptom is not considered prevalent or concerning in that disease. The reviewer questions specifically whether – in MGS – it is accurate to say that there are no affected immune cells. To date, there are three reports that specifically state that there are no abnormalities in immune cell populations and/or function (MCM5 patient, 1 ORC patient and 1 DONSON patient). Whereas only a few have reported on immune function, immunodeficiencies seen in the other diseases would have manifested in clinically relevant symptoms. With normal standards of care these symptoms would trigger clinical workup and we assume that abnormalities in the immune system would have been reported. Additionally, several patients were followed into adulthood so we can be confident that new immune defects or serious symptoms such as cancer do not occur later in life and are not reported due to lack of follow-up. We concede that some phenotypes may not be recognized and noted during patient care like dysmorphic facial features, however, the phenotypes that we discuss, such as immune deficiencies and skeletal anomalies, cause clinically relevant symptoms that suggest that specific cell types are affected by disease.

  1. Figure 1 and the model organism

When explaining the mechanism of DNA replication, the authors should specify the model organism (in the figure title/legend as well as in the main text) so as not to confuse the readers. Figure 1 basically describes our current knowledge of DNA replication initiation mechanism based largely on studies carried out in budding yeast but it shows human proteins. Nonetheless, at times the authors talk as if this figure is describing the budding yeast replication model and use the yeast gene nomenclature (line 115 paragraph, for instance). They provide some explanation in a paragraph that starts on line 150 but it seems a bit late and out of place.

We have addressed the reviewer’s point by moving line 150 sooner in the section (line 121) to introduce human proteins as we introduce their yeast counter parts. We agree that discussion of this is helpful to highlight differences between yeast and human systems.  Please see (lines 117-149). Additionally, we have added the sentences “Many of these findings are consistent with observations in human cells and we will note where the two systems diverge’ (lines 121-123) and “This cartoon depicts the steps and essential proteins involved in DNA replication origin licensing and firing drawing from knowledge gained by studying yeast and human systems’ (lines 97-98) to more clearly describe that much of the work to understand DNA replication has been performed in yeast, but that studies in human cells are consistent with these findings. Thus, in figure 1, we have depicted human proteins and interaction and filled in any gaps in knowledge with what is known from yeast systems. For instance, whereas Cdt1 binds Mcm2-7 in yeast, in humans CDT1 and MCM2-7 have not been isolated in complex together and are not depicted as such in our figure. Conversely, the studies on CMG bypass have been performed with yeast proteins and thus we relied on that data to generate our model.

  1. Termination

Replication termination is poorly discussed – only one short paragraph (line 189). For instance, they never talk about MiDAS.

Line 510–513: this seems not true. Two recent back-to-back Cell Research articles in 2020 have addressed this.

The reviewer notes that the discussion of termination is too brief. The rationale for this lies in the fact that no congenital diseases have been associated with proteins involved in termination. This has been further clarified by “Currently, there are no known congenital diseases caused by disruption of replication termination.” (line 213). Mitotic DNA synthesis (MiDAS) is generally not considered to be a part of normal DNA replication. MiDAS is a repair pathway that utilizes displacement synthesis, not a semi-conservative DNA replication mechanism. As discussed in the abstract we chose to focus this review only on DNA replication and not to include repair.

Lines 510-513 in the original version of our manuscript address chromosomal breakage in MCM4 mutants (“Regions that undergo breaks with aphidicolin treatment often correspond to common fragile sites, however follow-up studies to identify the chromosomal location of these breaks were not described [108] (now 110).”) Here, we meant to say that the exact break sites have not been mapped in MCM4 patients. We have revised the sentence to reflect this (line 543).

With regard to the papers the reviewer mentions, they seem to pertain to MiDAS, which is not included in this review because it is a repair pathway (see above).

  1. Lengthiness

- I felt that some parts of this review article focused too much on very basic textbook knowledge and history of a particular biological phenomenon and may not be totally necessary. At least, they could be shortened.

- There are unnecessary blank spaces here and there (p. 13, 17, 19, 20, 23, 26, 30, and on every page on the left side), which could be removed. These blanks also make it very difficult to read the manuscript.

- There are too many figures (12 figures) but I did not find Figs. 2–11 to be particularly informative. One could read the piece without going through the figures. Perhaps most of them could become Supplementary Figures.

As previously discussed, we wrote this review with the aim to be a resource for a broad audience and have provided the appropriate background to accommodate this audience. Additionally, the gene schematics we provide offer insight into location and clustering of mutations. The crystal structures were included for readers who do not routinely think about the structures of the complexes and how the various mutations might affect their formation. None of the information provided in the figures is currently available in textbooks.

Other points:

  1. Page numbers are messed up after page 17 (around line 630).
  2. The justification should be consistent throughout the manuscript. Currently, the text is full-justified except for lines 1055–1073, while the figure legends are left-justified.
  3. Line 124: ishi > ichi
  4. Line 359: CDC6 primary cilia > CDC6, primary cilia (comma)
  5. Lines 668, 838: unnecessary space
  6. References 16 and 17 are duplicated

Thank you, these have all been fixed.
